# Impact of anionic lipids on the energy landscape of conformational transition in anion exchanger 1 (AE1)

Tianle Chen [1,11], Francesca Vallese[2,3,4,5,11], Eva Gil-Iturbe [6], Kookjoo Kim[2,3], Tito Calì[7,8,9], Matthias Quick [3,6,10], Oliver B. Clarke [2,3] ✉ & Emad Tajkhorshid [1] ✉

Anion Exchanger 1 (AE1) is an elevator-type transporter that plays a key role in acid-base homeostasis of erythrocytes. Here, we report three high-resolution cryo-electron microscopy (cryo-EM) structures of distinct states of AE1: two inward-facing (IF1 and IF2) and one outward-facing (OF). Uptake assay revealed the modulatory effect of phosphatidylinositol 4,5-bisphosphate ($PIP_2$) lipids on AE1. Molecular dynamics simulations are conducted on these structures to determine the anion binding sites in AE1. We then use advanced enhanced sampling to study the OF⇌IF transition in AE1 in three systems: *apo*, $HCO_3^-$-bound, and an AE1 system in which cryo-EM-determined $PIP_2$ lipids had been removed. The transition pathways were then used to calculate the free energy of the OF⇌IF transition in AE1 under different conditions. The results show how substrate reduces the transition barrier against transport. Furthermore, they capture the inhibitory effect of $PIP_2$ lipids and provide a molecular mechanism for this inhibitory effect.

Anion exchanger 1 (AE1), *aka*, band 3, which is encoded by the SLC4A1 gene, is the most abundant membrane protein in human red blood cells (RBCs) accounting for ~25% of their total membrane proteins[1,2]. AE1 facilitates efficient transfer of carbon dioxide from tissues to the lungs by rapidly transporting the soluble bicarbonate ($HCO_3^-$) form across the RBC membranes[3]. AE1 mediates an electroneutral anion exchange by counter-transporting one $Cl^-$ ion against one $HCO_3^-$ during its transport cycle. This gradient-driven process results in a phenomenon known as the chloride shift, which enables RBCs to recognize metabolically active tissues and supply them with more oxygen[4,5]. AE1 mutations have been associated with various diseases including hereditary spherocytosis, Southeast Asia ovalocytosis, and distal renal tubular acidosis[6–8].

Human AE1 contains 911 residues which are divided into two domains: an N-terminal cytosolic domain (CD, residues 1–347) and a C-terminal transmembrane domain (TMD, residues 371–911), joined by a flexible linker (residues 348–370). The TMD is responsible for anion transport[9], while the CD mediates AE1's interaction with ankyrin, protein 4.2, deoxyhemoglobin, and glycolytic enzymes[10,11]. CD can be removed without affecting the transport function of AE1[12,13]. The TMD is composed of 14 transmembrane segments (TMs) which are organized into two inverted repeats of the same fold, TM1-7 and TM8-14[2]. These TM helices are arranged into a transport domain (TD: TM1-4 and TM8-11), also referred to as the core domain, and a scaffold domain (SD: TM5-7 and TM12-14), also known as the gate domain, with the

[1]Theoretical and Computational Biophysics Group, NIH Resource for Macromolecular Modeling and Visualization, Beckman Institute for Advanced Science and Technology, Department of Biochemistry, and Center for Biophysics and Quantitative Biology, University of Illinois, Urbana-Champaign, Urbana, IL, USA. [2]Department of Anesthesiology, Columbia University Irving Medical Center, New York, NY, USA. [3]Department of Physiology and Cellular Biophysics, Columbia University, New York, NY, USA. [4]Structural Biology Initiative, CUNY Advanced Science Research Center, New York, NY, USA. [5]Department of Chemistry and Biochemistry, City College of New York, New York, NY, USA. [6]Department of Psychiatry, Columbia University Irving Medical Center, New York, NY, USA. [7]Department of Biomedical Sciences, University of Padua, Padua, Italy. [8]Padua Neuroscience Center (PNC), University of Padua, Padua, Italy. [9]Study Center for Neurodegeneration (CESNE), University of Padua, Padua, Italy. [10]Area Neuroscience-Molecular Therapeutics, New York State Psychiatric Institute, New York, NY, USA. [11]These authors contributed equally: Tianle Chen, Francesca Vallese. ✉e-mail: oc2188@cumc.columbia.edu; emad@illinois.edu

anion translocation pathway located between the two domains. Similar to other facilitated transporters from the SLC4 family, e.g., the human electrogenic Na$^+$/HCO$_3^-$ cotransporter 1 (NBCe1) and the fungal borate transporter Bor1, AE1 operates via an elevator mechanism[14–16], in which the transporter switches between the inward-facing (IF) and outward-facing (OF) states by an elevator-like motion of the TD relative to SD. During this mechanism, the movement of the TD carries the substrate from one side of the membrane to the other. Recent structural studies have provided insights into substrate-interacting residues and inhibition of AE1, reported for the OF state[17–19]. There is however limited knowledge regarding its IF state[16,20] and the mechanism of its transition between the two states. While there is consensus on the alternating access mechanism of AE1[21], the structural transitions underlying the process have not been characterized.

Lipids are known to play crucial roles in transporters' function by influencing their structure and dynamics. For example, the effect of cholesterol on the ATP-binding cassette super-family G member 2 protein (ABCG2)[22] and PIP$_2$ (phosphatidylinositol 4,5-bisphosphate)-modulated activation of Na$^+$/H$^+$ exchangers[23] have been reported. Advances in cryo-EM have expanded the number of lipid-bound membrane protein structures, e.g., AE2 in complex with PIP$_2$[24], providing convincing evidence for their roles in transporter regulation. Enrichment and putative binding sites of PIP$_2$ and cholesterol have been also reported by recent computational studies, e.g., using coarse-grained simulations[25], but their potential functional roles require additional investigations.

In this work, we report cryo-EM structures of AE1 in both OF and IF states: one OF structure and two IF structures, IF1 and IF2, with different configurations of TM10. Arginine 730 (R730) is observed to be exposed to the lumen in both states, with MD simulations capturing frequent spontaneous binding of Cl$^-$ and HCO$_3^-$ to this residue. To investigate the OF⇌IF transition, we perform driven MD simulations along specific collective variables and computed the free energy profiles for the *apo*, substrate-bound, and PIP$_2$-depleted AE1 systems. These simulations reveal that substrate binding lowers the energetic barrier for the transition, thereby facilitating conformational change. Furthermore, PIP$_2$ binding sites are identified at the AE1 dimer interface, and functional uptake assays show an increased transport activity upon PIP$_2$ depletion. Simulations reveal that PIP$_2$ elevates the transition barrier and offer a mechanistic explanation for this inhibitory effect. Collectively, our findings provide a molecular-level understanding of the AE1 transport cycle and demonstrate the regulatory effect of PIP$_2$ on its activity.

## Results and Discussion

### Cryo-EM structures of AE1

Previously reported structures of AE1 have been solved in the OF conformation[16–18] except for a low-resolution structure of AE1 solubilized in n-dodecyl-β-D-maltoside where the transporter assumes an IF conformation[16]. Recently, a study was published by Su et al. in which they reported the first high resolution structure of AE1 dimer in the IF-OF state in the presence of bicarbonate[20]; the IF state of this structure corresponds to one of our IF states (IF1). To stabilize the IF state in native membranes, we incubated human erythrocyte ghost membranes with a membrane-permeable lysine-reactive crosslinker (DSP) before membrane solubilization, purification, and reconstitution in lipid nanodiscs. We purified the full-length AE1 dimer from human erythrocytes, including both the CD and the TMD. However, we were only able to resolve a high-resolution structure for the TMD part of the protein, as the CD is known to be highly mobile (Supplementary Fig. S1A). High-resolution visualization of CD is typically possible only when it binds to other proteins[10]. Using single particle cryo-EM, we were able to identify three different conformations for AE1 TMD using 3D classification: two IF states (IF1 and IF2) which are differentiated by the conformation of TM10 that bears the presumed anion-binding

residue R730, and the OF state (Fig. 1). AE1 monomers in the IF1 state represent 43.6%, the OF conformation accounts for 51.8%, and the IF2 state represents 4.6% of the total particles. In the IF1 state, TM10 is unwound and exhibits weaker density in this region; however, low-pass filtering can still reveal main-chain directionality up to residue 732, with R730 projecting into the inward-facing cleft—a structural feature consistent with the previously reported IF conformation[20] (Supplementary Fig. S2). In contrast, in the identified IF2 state, it is fully ordered, with R730 interacting with the N-terminal end of TM3 (Fig. 1C). Multiple different configurations of the AE1 dimer were separated by 3D classification, of which three major configurations were refined to high resolution: IF1/OF at 2.4 Å, OF/OF at 3.1 Å, and IF1/IF2 at 2.9 Å resolution.

As expected from prior studies[16–18], AE1 forms a homodimer. Analysis of different states within the same AE1 dimer clearly demonstrates that the two subunits function independently. The transport of Cl$^-$ and HCO$_3^-$ occurs at the interface between the SD and TD domains in each protomer. While SD seems stationary between the OF and IF states, significant motion is observed in TD, particularly in TM3 and TM10. These two helices are interrupted in the middle of the membrane, creating the appearance of a continuous helix at their juncture. The three different conformations of AE1 allow us to identify a set of critical residues potentially involved in ion access and binding. The opposite orientations of TM3 and TM10 lead to the formation of a putative, dipole-mediated, anion-binding site[2], which is alternately accessible to either side of the membrane in the OF and IF states through the elevator movement of the transporter's TD[26]. The TD undergoes the most notable change, particularly in the region that involves TM10. Residue R730, situated within TM10, occupies a crucial position within the putative anion cavity. Recent studies have demonstrated the involvement of this residue in coordinating HCO$_3^-$ within the binding pocket of AE1[16,20]. In the presence of 100 mM HCO$_3^-$, a cryo-EM study reported HCO$_3^-$ density near R730[17]. In the OF state, TM10 is well-ordered, with R730 facing the extracellular space. This state was previously analyzed for the human AE1 and showed an organization similar to other SLC4 transporters (e.g., NBCe1 and the Na$^+$-driven Cl$^-$/HCO$_3^-$ exchanger NDCBE)[14,27]. In the predominant IF state, IF1, TM10 is completely unfolded, with R730 inaccessible to the putative substrate pathway. We also observed a second IF state, IF2, in which R730 is directed towards the internal cavity and inaccessible from the extracellular space, with a displacement of only 7 Å compared to the OF state (Supplementary Fig. S3A). Meanwhile, TM3 maintains the same position in the IF1 and IF2 structures but shifts upward by 8 Å and rotates by 19° in the OF state (Supplementary Fig. S3B).

A comparison of the OF and IF conformations reveals that the transition between them occurs via TD's movement around an axis roughly intersecting with the intracellular distal corner (F507) (Supplementary Fig. S1B). This elevator-like motion is also reported for related transporters, e.g., SLC4A2[24]. Extra densities that correspond to phosphatidylcholine (PC) lipids (used for the nanodisc reconstitution) were modeled in the structure (Fig. 1A, B). We also modeled two PIP$_2$ lipids at the interface of the two AE1 subunits (Figs. 1 and 2).

### PIP$_2$ inhibition of AE1 activity

Lipid headgroup densities are seen at the intracellular dimer interface on the two sides of the protein, where they bind to patches of positively charged residues. Based on the distinctive shape of the densities, we have assigned them as PIP$_2^{[10]}$ (Fig. 2A and B). A series of well-ordered POPC (1-palmitoyl-2-oleoyl-glycero-3-phosphocholine) lipids is also present at the interface between the two protomers, which may contribute to dimer stabilization (Fig. 2C and D). Interfacial lipids have been implicated in maintaining the stability of related dimeric transporters, e.g., in the purine transporter UapA[28].

PIP$_2$ plays a critical role in RBC membrane signal transduction[29]. It modulates the interactions between the membrane cytoskeleton and

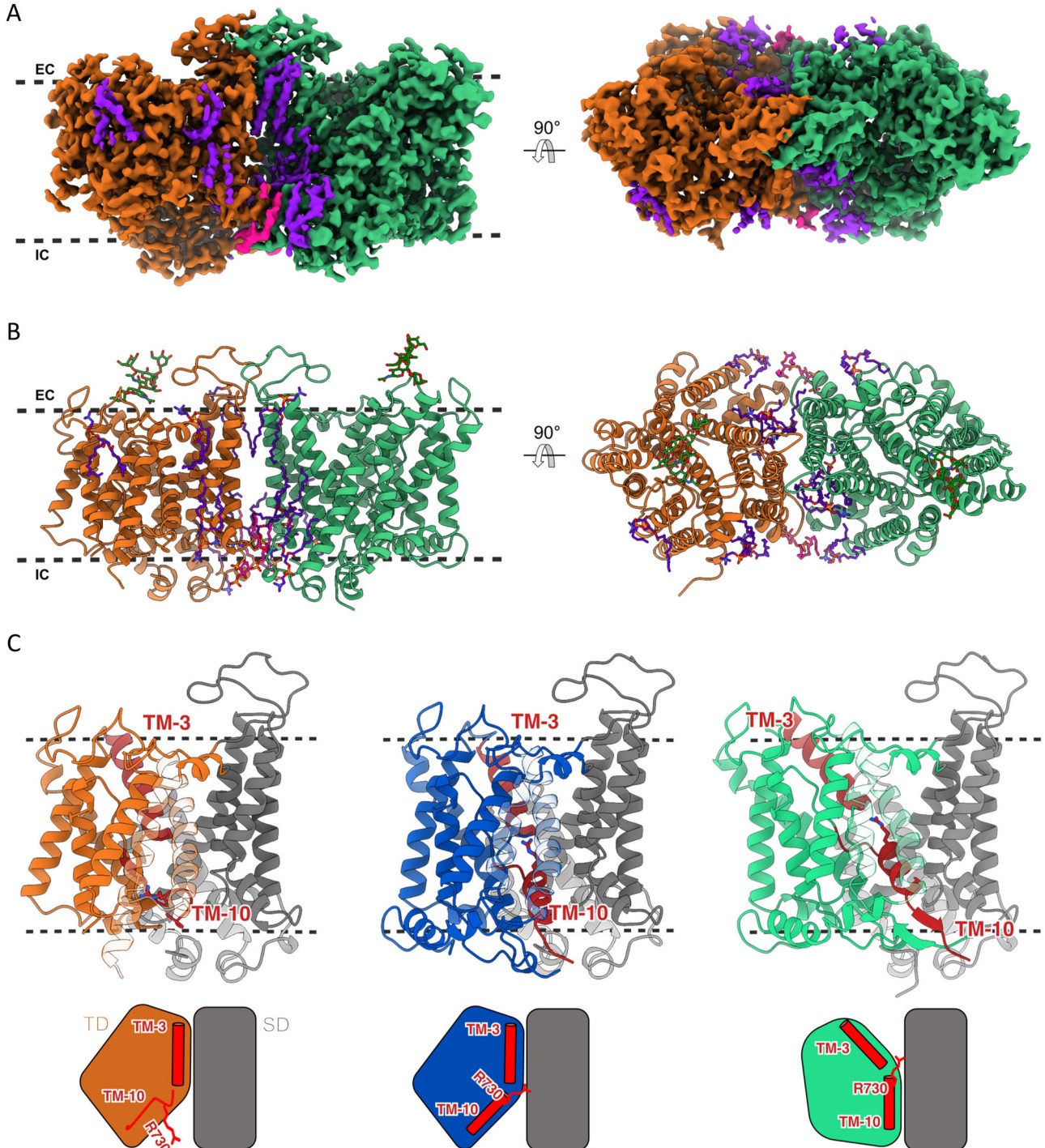

**Fig. 1 | Three conformations of human AE1 captured by Cryo-EM. A** The cryo-EM density map of AE1 TMD in IF1/OF configuration in the plane of the membrane (left panel) and viewed from the extracellular side (right panel). AE1 protomer in IF1 conformation is colored in orange, and the one in OF conformation is colored in green. Lipids are colored in purple, except for PIP₂, which is shown in pink. The black dotted lines represent the approximate boundaries of the cell membrane, with IC denoting the intracellular side and EC the extracellular side. **B** Model of AE1 dimer in IF1/OF conformation. Views and reference colors are the same as in A. The sticks in green and red represent N-linked glycosylation at residue N642 of each subunit. **C** AE1 protomer in IF1 state (orange/gray), IF2 (blue/gray) and OF state (light green/gray) with the corresponding schematic representation with TM-helices (TM3 and TM10) in red. The SD is displayed in gray, while the TD is colored according to the corresponding state.

the lipid bilayer, influencing cellular shape and deformability. The identification of this lipid at the AE1 dimeric interface from the cryoEM density between the two protomers (Fig. 2A and B) raises some questions about its possible role in the regulation of AE1. Several amino acids involved in PIP₂ binding are conserved within the SLC4 transporter family, particularly P815 and positively charged residues R602

and K817. R602 is located in the first helical turn of the N-terminal region of TM7, facing back toward the cytosolic loop connecting TM6 and TM7. When mutated to histidine, it causes recessive distal renal tubular acidosis[30]. This suggests that the positive charge at position 602 may be essential for stabilizing the AE1-PIP₂ interaction and, consequently, the stability of the dimer interface.

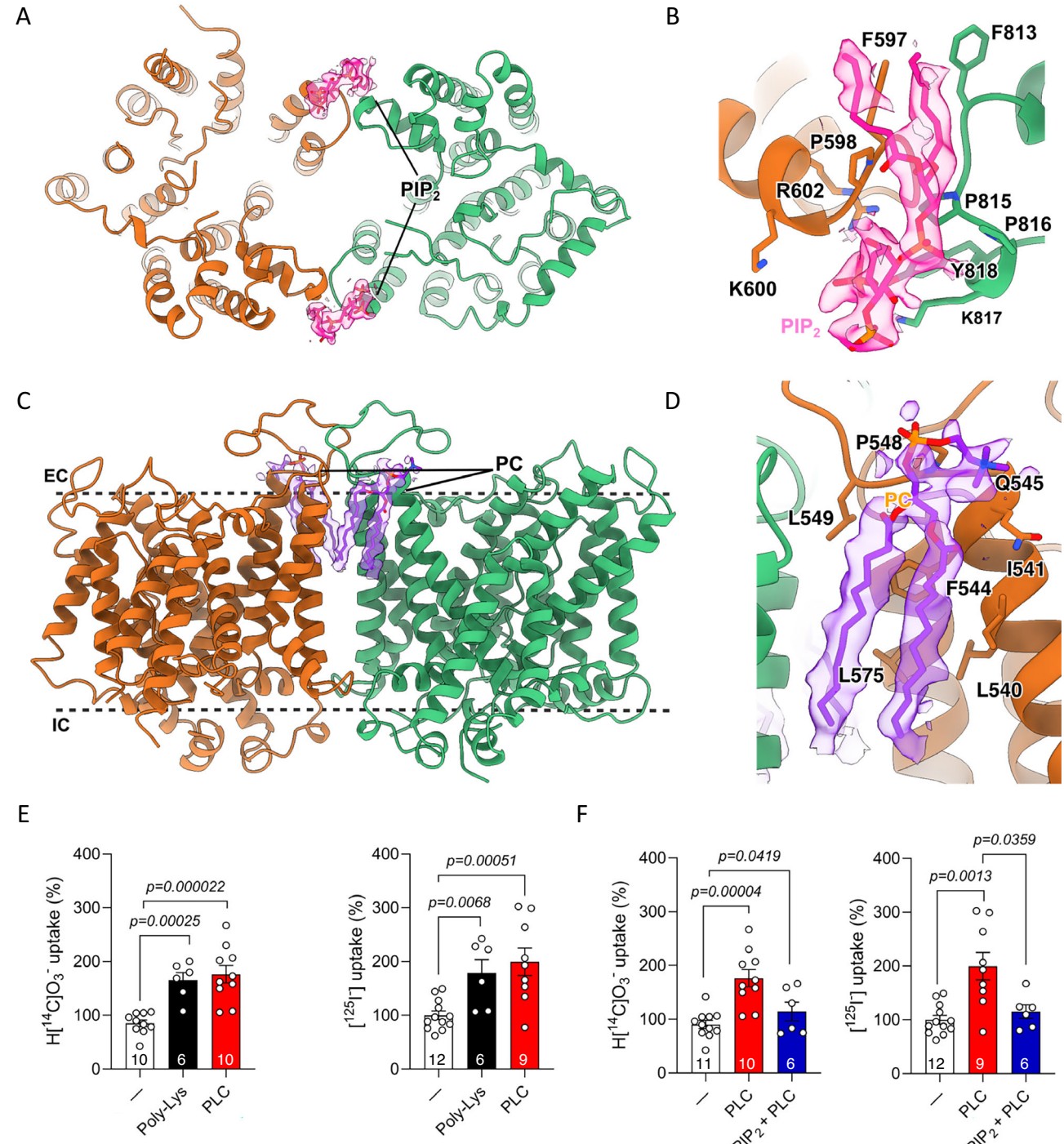

**Fig. 2 | AE1 in complex with PIP$_2$ and PC. A** Model of AE1 dimer as viewed from inside the cell. AE1 protomers are colored in orange (IF1) and light green (OF), with PIP$_2$ molecules in pink. **B** Close-up view of the AE1 dimer interface. PIP$_2$ (pink) sits in the middle of the site. The key residues that mediate the interactions with the lipid at the interfaces are shown in stick representation. **C** Model of AE1 dimer as viewed in the plane of the membrane. PC lipids are shown in purple. **D** Close-up view of the interactions between AE1 protomers and a PC lipid (purple). The key residues that mediate the interactions are shown in stick representation. **E**, **F** 1-min uptake of

10 μM NaH[$^{14}$C]O$_3$ or Na[$^{125}$I] by AE1-containing proteoliposomes. AE1 was incubated for 1 h with 15 mg/mL poly-lysine (Poly-Lys), 0.3 U/mL PLC before being incorporated into proteoliposomes. In F, 50 μM PIP2 was added to liposomes previously treated with PLC. Data in panels E and F are shown as mean value ± SEM from ≥ 6 biological replicates as indicated for each data set. The normality of the biological replicates was tested with the Shapiro–Wilk test, and a two-tailed Mann–Whitney U test was applied to non-parametric data, and a two-tailed unpaired Student's t-test was used for parametric data.

To test whether PIP$_2$ affects the activity of AE1, we first checked its possible effects on substrate transport using a proteoliposome-based assay. We reconstituted full-length AE1 into liposomes and measured the uptake of NaHCO$_3$ and NaI to assess the transport activity for the two of the established AE1 substrates[31]. We verified AE1-specific uptake

by performing the assays in the presence of DIDS (4,4'-Diisothiocyano-2,2'-stilbenedisulfonic acid), a known anion transporter inhibitor[32]. The uptake levels in the presence of DIDS were reduced by half compared to those without the inhibitor (Supplementary Fig. S1C). To determine AE1's affinity for HCO$_3^-$ and I$^-$, isotopic dilutions of the radiolabeled

anion with increasing concentrations of the unlabeled anion were performed (Supplementary Fig. S1D), yielding an EC50 (effective concentration of the unlabeled anion to reduce the uptake of radiolabeled anion by 50%) of 79 nM for $HCO_3^-$ and 143 nM for $I^-$.

To evaluate the effect of $PIP_2$ on AE1 activity, AE1's uptake activity in proteoliposomes was measured after preincubating it with Poly-Lys and phospholipase C (PLC), which cleaves phospholipids just before the phosphate group. Depletion of $PIP_2$ using PLC, or by using Poly-Lys, which shields $PIP_2$'s negative charges, activates AE1mediated $I^-$ or $HCO_3^-$ uptake (Fig. 2E). Notably, when $PIP_2$ is removed using PLC before its incorporation into proteoliposomes, an increase in substrate uptake is observed, which is then reversed once new $PIP_2$ is added (Fig. 2F). These results unequivocally demonstrate the involvement of $PIP_2$ lipids in regulation of AE1.

### Anion binding to AE1 different states

As an antiporter, AE1 relies on the coupling of downhill electrochemical gradient of one ion to transport another[33]. To gain insight into the binding of anions within the lumen formed between the SD and TD, we conducted 8 independent 1 μs simulations of the membrane-embedded TMD of AE1 using the IF1/OF and IF1/IF2 structures individually. AE1 dimer was embedded in a heterogeneous lipid bilayer and solvated with a solution of 18 mM $HCO_3^-$ and 87 mM $Cl^-$, approximating the average concentrations of these anions in the venous and arterial blood[34]. The hydration map of the lumen (Supplementary Fig. S4A and B) clearly demonstrates that in both the OF and IF states, the central lumen is exclusively exposed to only one side of the membrane (alternating access) from where the anions may diffuse into and out of the protein. In the IF1 state, TM10 is unfolded and the putative substrate-binding residues fail to form a well-defined binding site; our simulations also show that anions merely remain near the intracellular vestibule. This observation is consistent with a recent IF structure in which no bound ion was detected[20]. In contrast, in our IF2 state, the well-ordered TM10 appears to form a structured binding site for anions, as corroborated by our simulation data indicating preferential anion binding. Consequently, we focused our further analysis exclusively on the OF and IF2 protomers.

Over the collective course of 8 μs sampling, recurring cases of $Cl^-$ or $HCO_3^-$ binding/unbinding to/from the protein lumen were observed for both OF and IF2 states. One such $HCO_3^-$ binding event is highlighted in Supplementary Movie 1. Averaged over the simulations, the spatial distributions of the anions in the vicinity of AE1 are shown as volumetric maps in Fig. 3A and B, along with their probability density projected over the membrane normal (z-axis) (Supplementary Fig. S4C). The luminal anion distributions within the OF and IF2 states are distinct, with the probabilities exhibiting two different peaks around $z = -5$ and $z = 5$ Å, respectively. These regions are both accessible to R730 in the OF and IF2 states. A high-density anion site is also observed near the cytoplasmic opening in both OF and IF states (localized at $-15 \leq z \leq -10$ Å) where multiple positive residues from TM6 and TM7 are located, including K590, K592, K600, R602, and R603. These residues clearly play a role in attracting and recruiting free anions from the solution.

To further characterize the anion binding sites in AE1, we ranked the luminal residues interacting with the anions during the simulations. The analysis revealed that, in the OF state, R730 consistently forms direct contacts with the anions (Fig. 3C). Additionally, we found that E535 coordinates with the bound bicarbonate through a hydrogen bond with the anion's hydroxyl group in ~70% of the anion-binding cases. In the IF state, R730 and R589 are the top anion-binding residues, both with a probability of ~50% for directly contacting the bound anions (Fig. 3D). R589 is positioned deep in the lumen and close to R730, together forming a positive site, where the anions can shuttle between them.

These findings are largely consistent with recent computational and cryo-EM studies[16,17]. The simulations clearly depict binding events where $HCO_3^-$ and $Cl^-$ ions enter and remain within the lumen, establishing stable interactions with the key residue, R730. In addition, we identify other luminal residues that contribute to AE1 anion binding.

Next, in order to obtain information on the kinetics of anion binding, we approximated the binding affinities for $Cl^-$ and $HCO_3^-$ based on the residence time of the anion (directly interacting with luminal residues) (Fig. 3E, F, G and H). Our calculations yield estimates for the dissociation constant $K_d$ of 255 mM for $Cl^-$ and 22 mM for $HCO_3^-$ in the OF state. For the IF state, the $K_d$ values are estimated to be larger, 364 mM for $Cl^-$ and 38 mM for $HCO_3^-$. These values are relatively higher, e.g., compared to an NMR estimate of 5.4 mM for $HCO_3^{-35}$. This discrepancy likely arises from the limited simulation timescale and the resulting small sample size for the binding events, leading to overestimated unbinding times due to diffusion effects and underestimated binding times.

### Constructing the OF⇌IF transition pathway

Central to the transport reaction catalyzed by antiporters such as AE1, is the conformational changes that bring about the alternating access mechanism[36,37]. To characterize the underlying structural basis for this mechanism and how $PIP_2$ lipids might influence the transition, we employed enhanced sampling techniques to reconstruct the transition in three systems: (1) substrate-free (apo) AE1, (2) a substrate-bound state (with $HCO_3^-$) identified during our spontaneous ion binding simulations, and, (3) a $HCO_3^-$-bound model in which the cryo-EM resolved $PIP_2$ lipids at the interface were removed. Conformational transition from the OF to the IF2 state was realized using driven simulations employing two CVs[38] designed based on our previous experience with elevator transporters[39,40], namely, translation (z) and rotation (θ), which represent the distance between the centers of masses (COMs) of the TD helices in the two states, and their relative orientation, respectively (Fig. 4A). These CVs effectively cover the translation and reorientation of the TD with respect to the SD, which are the major structural differences between the OF and IF2 states.

Additionally, we explored three protocols of the two CVs to optimize the biasing protocol: inducing rotation first and translation next, applying them in the reverse order, or simultaneously driving both CVs[41] (Supplementary Fig. S5A). The non-equilibrium works accumulated during these exploratory simulations indicated that the most efficient approach was simultaneously driving the two CVs (Supplementary Fig. S5B). In all three systems, the same protocol was applied when inducing the state transition. The Cα root mean square deviation (RMSD) of the helices with respect to the target IF state decreases from 4 Å to approximately 1 Å and remains stable throughout the post-pulling, free equilibration stage (Fig. 4B and Supplementary Fig. S6A and B). The CV values also reach their target values and remain there even after the restraints are fully released, indicating that the protein has successfully transitioned to a stable IF state (Fig. 4C, D and Supplementary Fig. S6).

Once the system reaches a stable IF conformation, and restraints used to maintain the substrate in the binding pocket during the transition are removed, $HCO_3^-$ spontaneously diffuses out of the protein lumen into the cytoplasm during the final equilibration stage, thereby completing its transport across the membrane (Supplementary Fig. S7A and C). Additional SMD simulations driving the reverse transition from the IF2 to OF state demonstrate that the $HCO_3^-$ transport in the opposite direction (Supplementary Fig. S7B). The induced conformational changes and the substrate transport are demonstrated in Supplementary Movie 1. This observation is yet another indication that the cytoplasmic gate had entirely opened (alternating access) in the final IF state, allowing the substrate to be released into the solution.

Since the initial estimate of the transition pathway is obtained from driven simulations, it is necessary to refine it into a more accurate

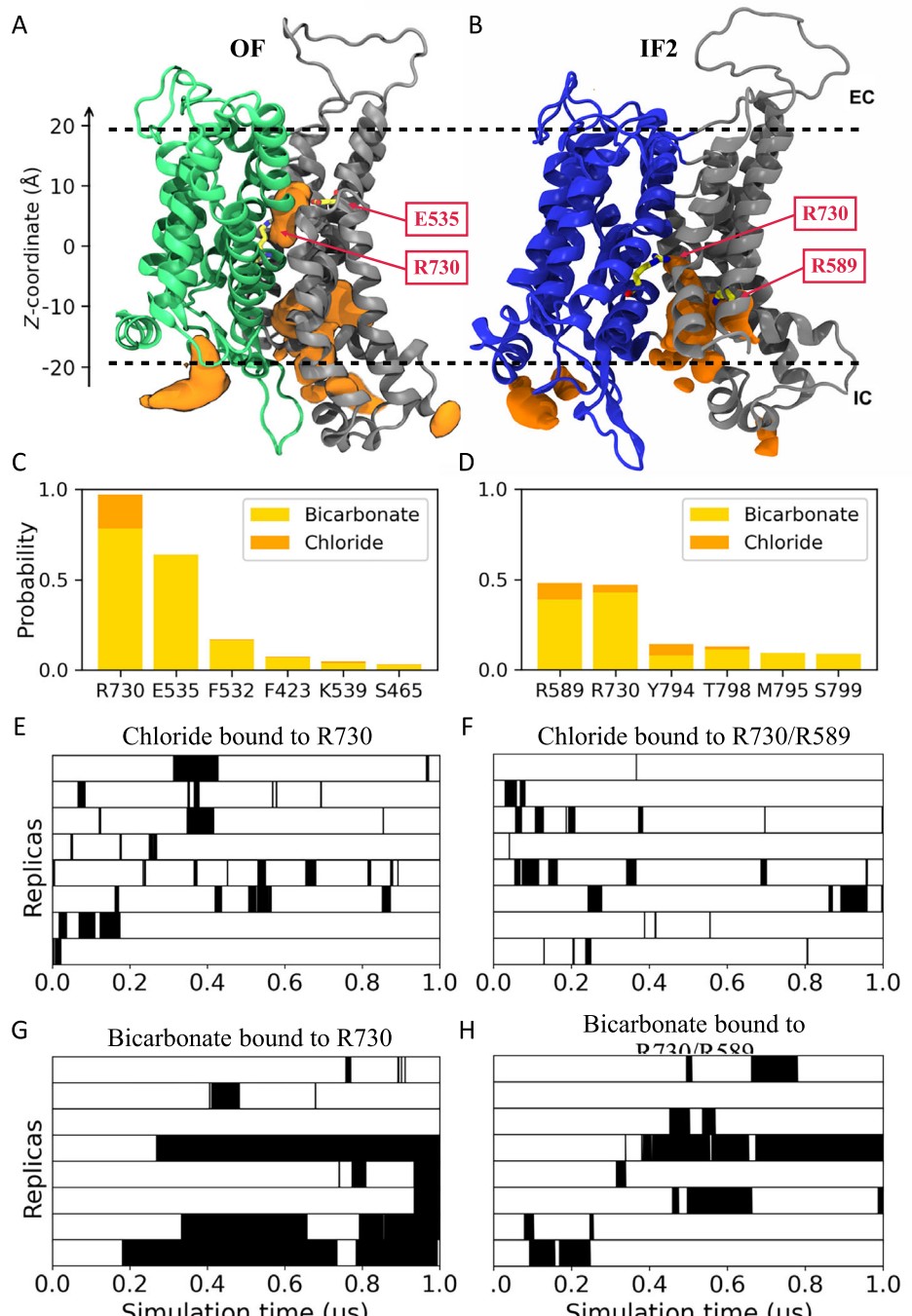

**Fig. 3 | Spontaneous anion binding to AE1 captured in MD simulations. A**, **B** The anion spatial distribution in the OF **A** and IF **B** states. The TD is shown in green and the SD in gray. The regions for anion binding are shown as orange volumetric surfaces. The main anion-interacting residues are shown in stick representation and labeled. **C**, **D** Probability for major luminal residue to interact with anions obtained from binding events in the OF **C** and IF **D** states. For each residue in the diagram, interactions with $Cl^-$ are represented in orange, and interactions with $HCO_3^-$ are shown in yellow. **E**, **F** $Cl^-$ interactions in the OF **E** and IF **F** states. The black regions represent the time intervals during which anions are bound. **G**, **H** $HCO_3^-$ interactions in the OF **G** and IF **H** states during the simulations.

one. The string method with swarms of trajectories (SMwST; see Methods) was utilized for this purpose[42]. After 200 iterations of SMwST simulations for multiple replicas, involving a cumulative 3μs of sampling for each system, the strings in the CV space drift from the original trajectory and eventually converge (Supplementary Fig. S8A, B and C). A comparison of the CV changes among the three systems reveals that the presence of $HCO_3^-$ induces subtle deviations in the mid-region of the movement profile of the TD, while the absence of $PIP_2$ leads to a slight reduction in the TD's translational displacement (Fig. 4E). These findings suggest that both $HCO_3^-$ and $PIP_2$ exert

modest influences on the transition profile of TD, although their overall impact remains minimal.

To gain insight into the thermodynamics of the transport and how substrate binding might impact the free energy of the conformational changes during the transport cycle, the refined transition pathways are used to calculate the free energy profiles using bias-exchange umbrella sampling (BEUS) simulations[43]. The free energy profiles for the OF⇌IF transition (Fig. 4F) obtained from reweighting the sampling feature an energy barrier separating the OF and IF states. Notably, the IF state occupies a higher free energy basin compared to the OF state. The

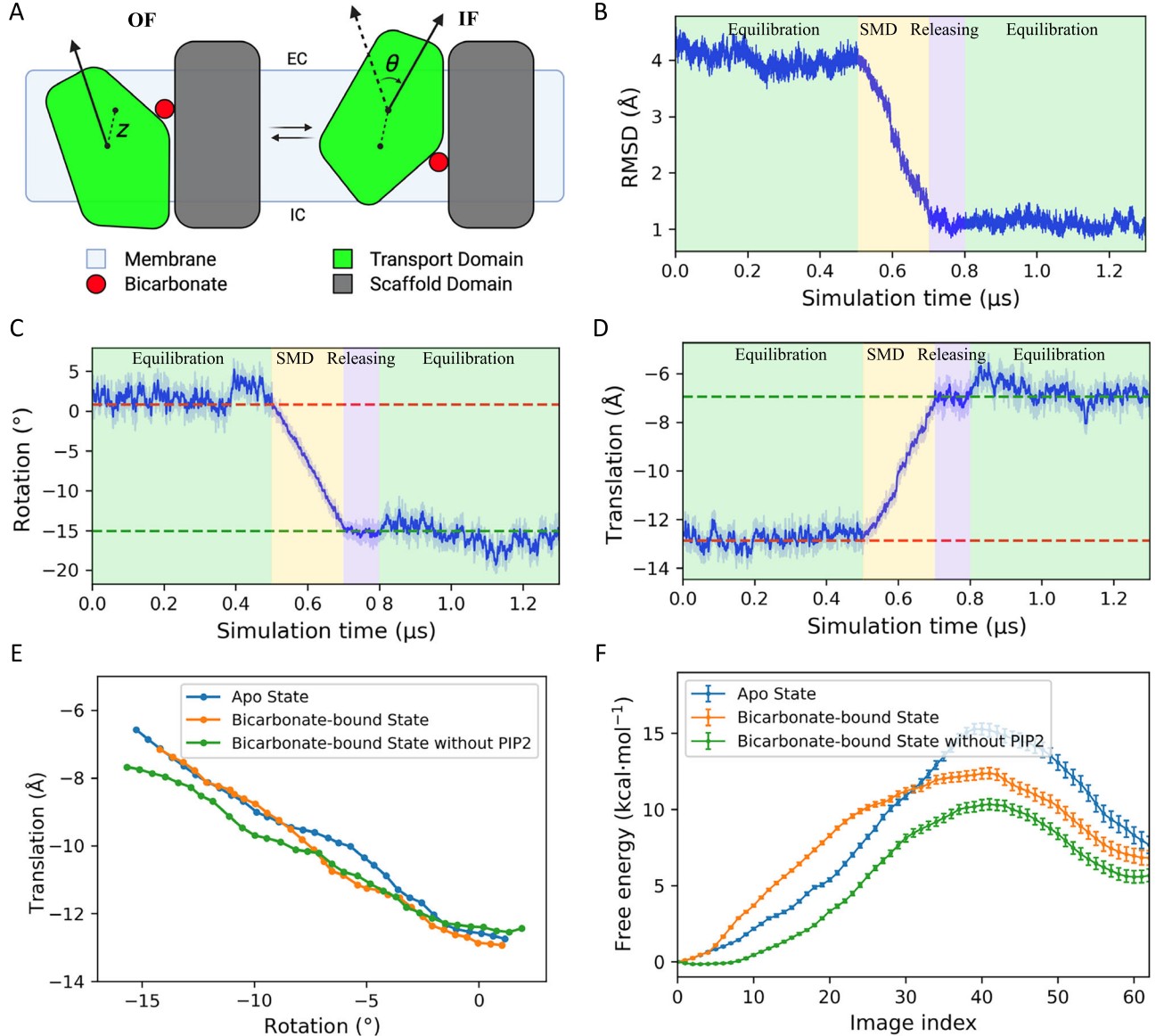

**Fig. 4 | Conformational transition of AE1 between the OF and IF states.**
**A** Schematic of the elevator mechanism for the OF ⇌ IF transition. The definition of the two CVs is shown, where $z$ and $\theta$ represent the translational and rotational motions of the TD relative to the SD, respectively. **B** Cα RMSD of the TM helices during the driven simulations with respect to the target structure in the IF state with SMD denotes the pulling stage. **C**, **D** The change of the CVs for the rotation **C** and translation **D** during the initial equilibration, driven (SMD), releasing, and final equilibration simulations, highlighted with different background colors. The CV

values of initial and target structures are shown as the dashed red and green lines, respectively. **E** Refined transition pathways with SMwST for the *apo* (blue), substrate-bound (orange), and PIP₂-removed (green) systems. Each point represents the averaged centers of 8 replicas for 32 swarms. **F** Free energy profiles computed using BEUS for the *apo*, substrate-bound, and PIP₂-removed systems with same colors as in **E**. Data are presented as mean ± SD, calculated via bootstrap resampling across 64 windows.

elevated free energy of the IF state could explain its lower prevalence in previous structural studies[10,17–19]. The similar amount of particles in the OF or IF states observed in our structure is likely due to the incubation of the ghost membrane with the DSP crosslinker, which facilitates the crosslinking of lysine residues (K539 and K851) when the transporter adopts an inward-facing conformation. Once crosslinked, this modification becomes irreversible.

**Effect of substrate binding on the free energy profile**
Comparing the free energy profiles of the substrate-bound and *apo* forms of AE1 shows a clear preference for the conformational transition when the substrate is bound, as indicated by a lower barrier by approximately 3 kcal/mol (Fig. 4F). This explains why substrate binding facilitates the global transition in AE1 as an antiporter. The

rotation and displacement of TM3 (residues 466–482) and TM10 (residues 728–739), which form the binding site, along the refined trajectories were calculated by averaging across replicas for each SMwST window. The movement of TM3 remains consistent across different systems (Supplementary Fig. S3C), while for TM10, the presence of the substrate induces an earlier translation with reduced rotation compared to the *apo* system (Supplementary Fig. S3D).

To further explore the structural changes occurring during the OF⇌IF transition and the effect of substrate binding, specific residue-residue interactions near R730 and at the domain interface are analyzed. The region around R730 displays different intermediates during the transition in the *apo* and substrate-bound systems (Fig. 5). For example, the space formed by the following two distances can

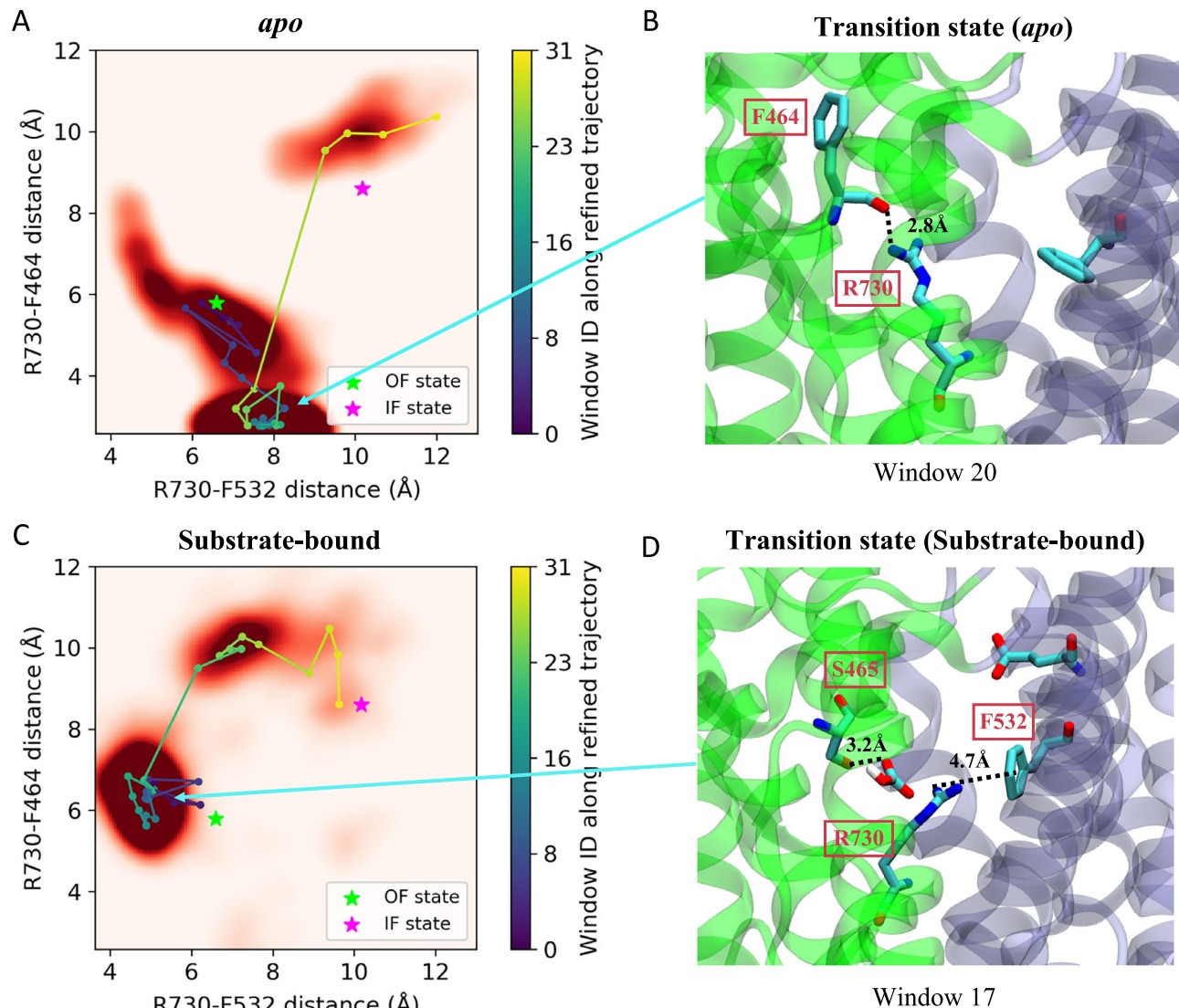

**Fig. 5 | Effect of substrate binding on the transition. A** Distribution of R730-F532 and R730-F464 distances during SMwST simulations for the *apo* system. The darkness of the red color indicates the density of sampling. Stars denote the locations of the OF and IF states, averaged from the equilibrium simulations. Colored data points represent average values per SMwST window (0 to 31; scale bar), with connecting lines indicating the refined transition pathway from OF to IF. **B** Molecular view of the transition state in the *apo* system from SMwST window 20 as an example, where the R730 interacts with the F464 backbone. **C** Distribution of R730-F532 and R730-F464 distances in the substrate-bound system. **D** Molecular view of the transition state in the substrate-bound system SMwST window 17, highlighting the newly formed R730's contact with F532 due to the hydrogen bond formed between $HCO_3^-$ and S465 (see text).

demonstrate the mechanistic differences: the distances between the R730 guanidinium group to the F532 aromatic ring and to the F464 backbone oxygen. These distances are monitored along the refined pathways and averaged over multiple replicas for each point on the string (Fig. 5).

In both end states (OF and IF states), R730 is relatively free and does not engage in any stable interactions, resulting in significant fluctuations of distances involving R730. In contrast, in the *apo* transition state, R730 forms a hydrogen bond with the backbone of F464 (Fig. 5B). They stay mostly within 2.6-3.0 Å during transition (windows 10-26), and thus R730 is kept far from F532 in the SD. During the transition in the substrate-bound protein, and as the TD moves towards the intracellular side, the $HCO_3^-$ is translocated deeper into the central lumen and comes in contact with the hydroxyl group of S465, establishing a hydrogen bond (Fig. 5D). The steric effect of the substrate alters the conformation of R730 causing it to establish a contact with F532 in the SD, which we believe is responsible for stabilizing the transition state and lowering the energy barrier. Such

cation-π interactions have been shown to be responsible for stabilizing both the overall protein structure and specific conformational states[44–46]. This observation is in line with the previous mutagenesis study where mutating S465 significantly reduced the activity but mutation of F464 showed no effect[47].

**Impact of PIP$_2$ on the transition**

To evaluate the stability of the cryo-EM resolved lipids during the equilibrium simulation, we analyzed the RMSD of three distinct lipid groups: (1) peripheral POPC lipids, which are located on the surface of the protein, (2) POPC lipids at the dimer interface, and (3) PIP$_2$ lipids. Supplementary Fig. S9 presents the headgroup RMSD values of these lipids relative to their initial positions, along with their standard deviations. Among these groups, PIP$_2$ lipids exhibit the lowest RMSD values and minimal variability, indicating greater stability even compared to POPC lipids confined within the dimer interface. This observation suggests that PIP$_2$ lipids are more likely to play a structural or functional role in the AE1 system. The interactions of the PIP$_2$

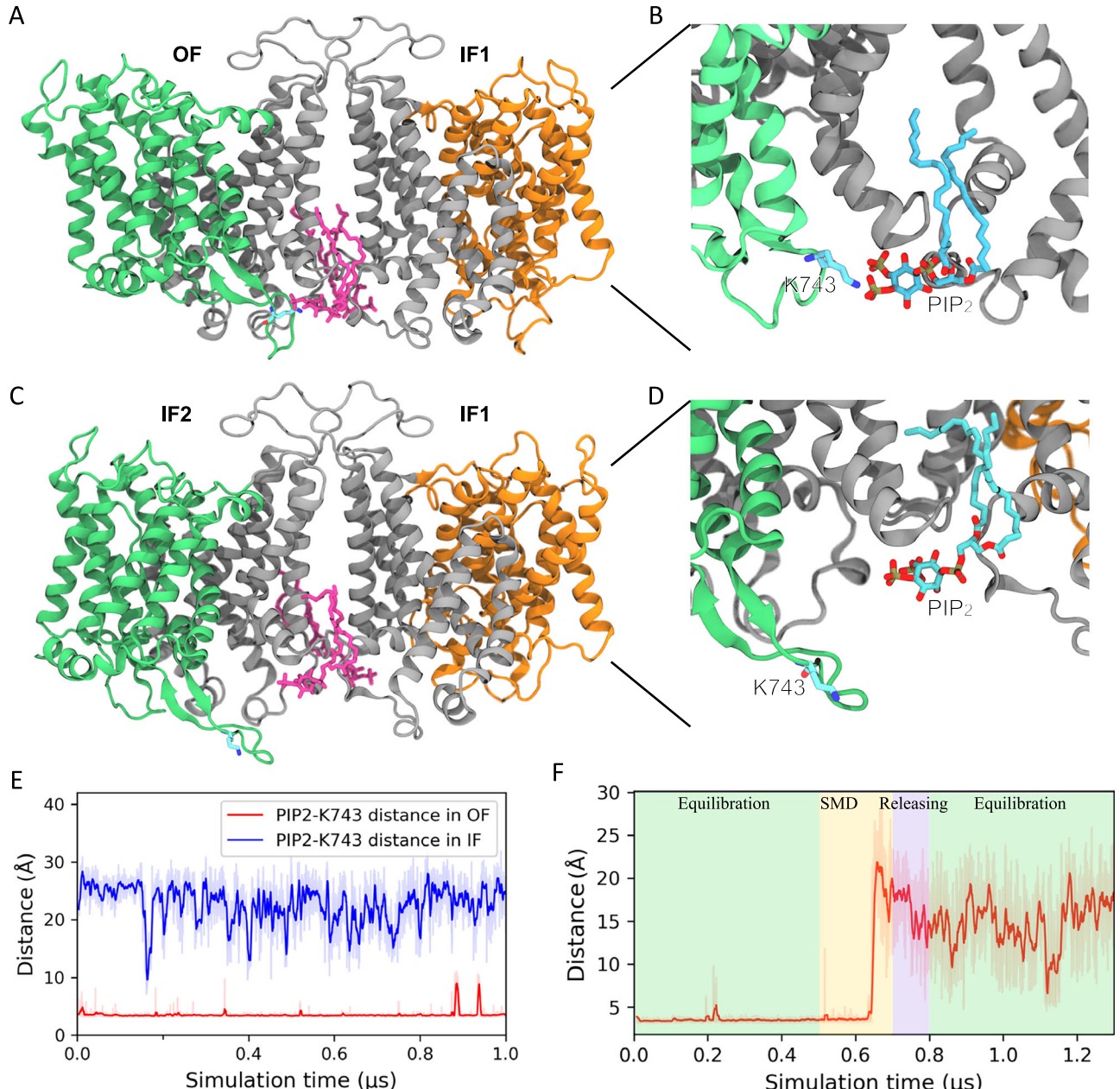

**Fig. 6 | Mechanism of PIP$_2$ effect on AE1 transition. A** Molecular representation of the PIP$_2$-β-hairpin interaction in the OF state (green/gray), highlighting PIP$_2$ (pink) and K743 on the loop (stick representation). **B** Enlarged view showing the interaction of K743 and PIP$_2$ in the OF state. **C** Molecular representation of AE1 after it transitions to the IF state. **D** Enlarged view showing the breaking of the interaction between K743 and PIP$_2$ in the IF state. **E** Time series of the shortest heavy atom distances between PIP$_2$ and K743 during the 1 μs equilibrium simulation for the OF (red) and IF (blue) states. **F** Time evolution of the PIP$_2$-K743 distance when the protein transitions from the OF to IF state.

headgroup with K600, R602, and R603 in the SD appear to be important elements for stabilizing the lipid's binding pose. Although PIP$_2$ lipids are situated between the two SDs, a notable difference between the OF and IF states is the interaction of one of them with an intracellular β-hairpin loop from the TD between TM10 and TM11 (Fig. 6A, C). In the OF state, K743 in this hairpin forms a salt bridge with the 4- or 5-phosphate groups in the PIP$_2$ headgroup, resulting in its structural stability throughout the 1 μs simulation time (Fig. 6B, E). This contact needs to be broken before reaching the IF state, in which K743 and the lipid are separated by 20-30 Å (Fig. 6D, E). Notably, the loop and PIP$_2$ are far from each other in both IF1 and IF2 states. Similar interactions between PIP$_2$ and the β-hairpin were also highlighted in previous coarse-grained simulations[25].

As expected, a sudden dissociation of this salt bridge is observed during the driven transition from the OF to the IF state (Fig. 6F and Supplementary Movie 1). Note that only the TM helices are biased during these simulations, and the β-hairpin loop is free of any restraints/biases during this phase. The differences observed in transitions among refined trajectories further highlight the impact of PIP$_2$ on AE1 activity. The removal of PIP$_2$ likely reduces the confinement of TM10 rotation, allowing it to rotate more extensively alongside its translation compared to the HCO$_3^-$-bound system, while still showing earlier translational movement than the *apo* system (Supplementary Fig. S3D). BEUS calculations show that the free energy barrier for the OF-to-IF transition in the PIP$_2$-removed system is approximately 2 kcal/mol lower than that of the native state (Fig. 4F), indicating an increased

rate of conformational change. This finding aligns with our uptake assay, which shows that the removal of PIP$_2$ increases AE1's activity. A previous study on bicarbonate transporter-related protein-1 (BTR1) also demonstrated that disruption of the PIP$_2$ binding site can shift the transporter into an IF conformation[48]. These results explain our experimentally measured inhibitory effect of PIP$_2$ on AE1 activity, highlighting the potential functional role of the TM10-TM11 loop in the transporter's function.

## Methods

### AE1 sample preparation for cryo-EM

Erythrocytes from fresh human blood samples were washed twice in 5 volumes of 130 mM KCl, 10 mM Tris-HCl, pH 7.4. The cells were hemolyzed in 5 volumes of 1 mM EDTA, 10 mM Tris-HCl, pH 7.4, and centrifuged at 18,000 × g for 10 min. The ghost membranes were then washed five times in the hemolysis buffer, and four additional times in 10 mM HEPES, pH 7.4. The hemoglobin-free ghost membranes were finally resuspended in 130 mM NaCl, 20 mM HEPES, pH 7.4, 0.5 mM MgCl$_2$, 0.05 mM CaCl$_2$, 2 mM dithiothreitol (DTT), and stored at –80 °C.

Erythrocyte ghost membranes were prepared as described by Niggli et al. 2016[49]. Ghost membranes (~3 g) were incubated with 2.5 mM DSP (dithiobis(succinimidyl propionate)) crosslinker in 10 mM HEPES (pH 7.4), 130 mM KCl at room temperature for 30 min. The cross-linking reaction was quenched by the addition of 20 mM Tris-HCl, pH 8. Membranes were washed 3 times with low-salt buffer (10 mM HEPES, 0.1 M KCl, 0.5 mM DTT, 0.5% Triton X-100, and 1 mM EDTA, pH 7.4), followed by incubation in high-salt buffer (10 mM HEPES, 1 M KCl, 0.5 mM DTT, and 0.5% Triton X-100, pH 7.5 and protease inhibitors) on ice for 30 min. The insolubilized material was pelleted by centrifugation (100k × g, 30 min), and the supernatant containing solubilized proteins was concentrated using a 100 kDa centrifugal concentrator (Millipore). The sample was then loaded on a glycerol gradient (12–28% glycerol in 10 mM HEPES, 130 mM NaCl and 0.05% Triton X-100, pH 7.5), and centrifuged for 15 h at 92,000 × g (Beckman SW 32 rotor). The fractions enriched with AE1 protein were collected and incorporated into lipid nanodiscs with a 1:5:250 molar ratio of protein, membrane scaffold protein (MSP-1E3D1), and POPC (Avanti polar lipids). The protein in MSP nanodisc was then subjected to a first gel filtration run using a Superdex 200 10/300 Increase column (Cytiva) and the main peak, close to the void volume of the column, was collected and further purified using a Superose 6 Increase 10-300 column (Cytiva) (Supplementary Fig. S10). 3 μL of purified AE1 at 5 mg/mL was added to a glow discharged (PELCO easiGlow) 0.6/1 μm holey gold grid (Quantifoil UltrAuFoil) and blotted for 4-6 s at 4 °C and 100% humidity using the Vitrobot Mark IV system (Thermofisher Scientific), before plunging immediately into liquid ethane for vitrification. The cryo-EM data were collected on a Titan Krios electron microscope (Thermofisher Scientific) equipped with a K3 direct electron detector (Gatan) operating at 0.83 Å per pixel in counting mode using Leginon automated data collection software[50]. Data collection was performed using a dose of 58 $e^-$/Å$^2$ across 50 frames (50 ms per frame) at a dose rate of 16 $e^-$/pixel/s, using a set defocus range of –0.5 to –1.5 μm. A 100-μm objective aperture was used. A total of 20,464 micrographs were collected.

### Cryo-EM data processing

The final cryo-EM data processing workflow is summarized in Supplementary Fig. S11. Orientation distributions, FSC plots, and validation statistics are presented in Supplementary Figs. S12, S13, S14, and Table S1. The map-to-model fits of TM7 and 10 for all three states are also presented in Supplementary Fig. S15. Maps have been deposited at EMDB (IF1-OF EMD-48421, OF-OF EMD-48422, IF1-IF2 EMD-48480).

Patch-based motion correction and doseweighting of 20 k movies were carried out in cryoSPARC using the Patch Motion job type. Subsequent steps were performed in cryoSPARC v3.2 and v3.3 unless otherwise indicated[51]. Patch-based CTF estimation was performed on the aligned, non-dose-weighted averages using Patch CTF. The particle picking was performed using crYOLO 1.8[52], and then coordinates were imported into cryoSPARC. The initial set of 4,942,577 particles was imported for extraction, ab initio reconstruction, and heterogeneous refinement, which resulted in 1,390,266 particles. Subsequent non-uniform refinement of the selected particles resulted in a map at 3.02 Å. Three rounds of Bayesian polishing of this set of particles were executed in RELION 4[53]. Between each two rounds of polishing, we imported the polished particles into cryoSPARC, followed by nonuniform refinement including on-the-fly per particle defocus refinement and refinement of higher-order CTF parameters (beam tilt and trefoil). The final round of polishing gave a 2.45 Å resolution reconstruction, with significantly improved density quality.

The polished particles were subjected to heterogeneous refinement using four AE1 classes as inputs, as well as three decoy classes, using a batch size per class of 10,000 particles. After heterogeneous refinement, the particles in the four AE1 classes (1,286,208 particles) were combined and used for 3D classification without alignments. 3D classification without alignments, requesting 50 classes, allowed the identification of 3 conformations for the AE1 monomer, corresponding to the IF1, IF2, and OF states. Combining different classes containing these conformations gave three maps – the IF1-OF state at 2.4 Å (666 k particles), the OF-OF state at 3.11 Å (109 k particles), and the IF1-IF2 state at 2.88 Å resolution (79 k particles). Atomic models of the three different conformations were built in Coot 0.9.8.96[54,55] using AE1 in the OF state (PDB:4YZF) as a starting model, followed by refinement using phenix.real_space_refine (PMID: 29872004). Model and density visualizations were prepared using ChimeraX[56].

### Mass Photometry (MP, iSCAMS)

Mass photometry experiments were performed with a Refeyn OneMP (Refeyn Ltd.). Data acquisition was performed using AcquireMP (Refeyn Ltd. 172 v2.3). Samples were evaluated with microscope coverslips (cover glass thickness 1.5 × 24 × 50 mm, Corning). The coverslips were washed with ddH$_2$O and isopropanol. A silicone template was placed on top of the coverslip to form reaction chambers immediately prior to measurement. The instrument was calibrated using NativeMark Protein Standard (Thermo Fisher). 10 μL of fresh room-temperature buffer was pipetted into a well, and the focal position was identified and locked. For each acquisition 1 μL of the protein (at a concentration of 200 nM) was added to the well and thoroughly mixed. MP signals were recorded for 60 s to allow the detection of at least $2 \times 10^3$ individual protein molecules. The data were analyzed using the DiscoverMP software.

### Proteoliposome preparation

Purified AE1 was reconstituted into preformed liposomes composed of POPC at a protein-to-lipid ratio of 1:150 (w/w) following a previously described protocol[57]. The lumen of the proteoliposomes was composed of 50 mM KPi, pH 7.5 and 10 mM NaSO$_4$ to promote the exchange with the substrates. To maintain their stability, the proteoliposomes were aliquoted, rapidly frozen in liquid nitrogen, and stored at –80 °C. In all experimental setups, liposomes with the same composition but lacking the AE1 protein were used as control.

### Liposome-based, radiolabeled uptake assays

Uptake of 10 μM Na[$^{125}$I] (0.5 mCi/mmol) and NaH[$^{14}$C]O$_3$ (58 mCi/mmol, both American Radiolabeled Chemicals, Inc.) was performed in AE1-containing proteoliposomes (~50 ng AE1 per transport assay) in 50 μL assay buffer composed of 110 mM KPi, pH 7.5 and 10 mM NaSO$_4$ over 10-second periods in the presence or absence of compounds as

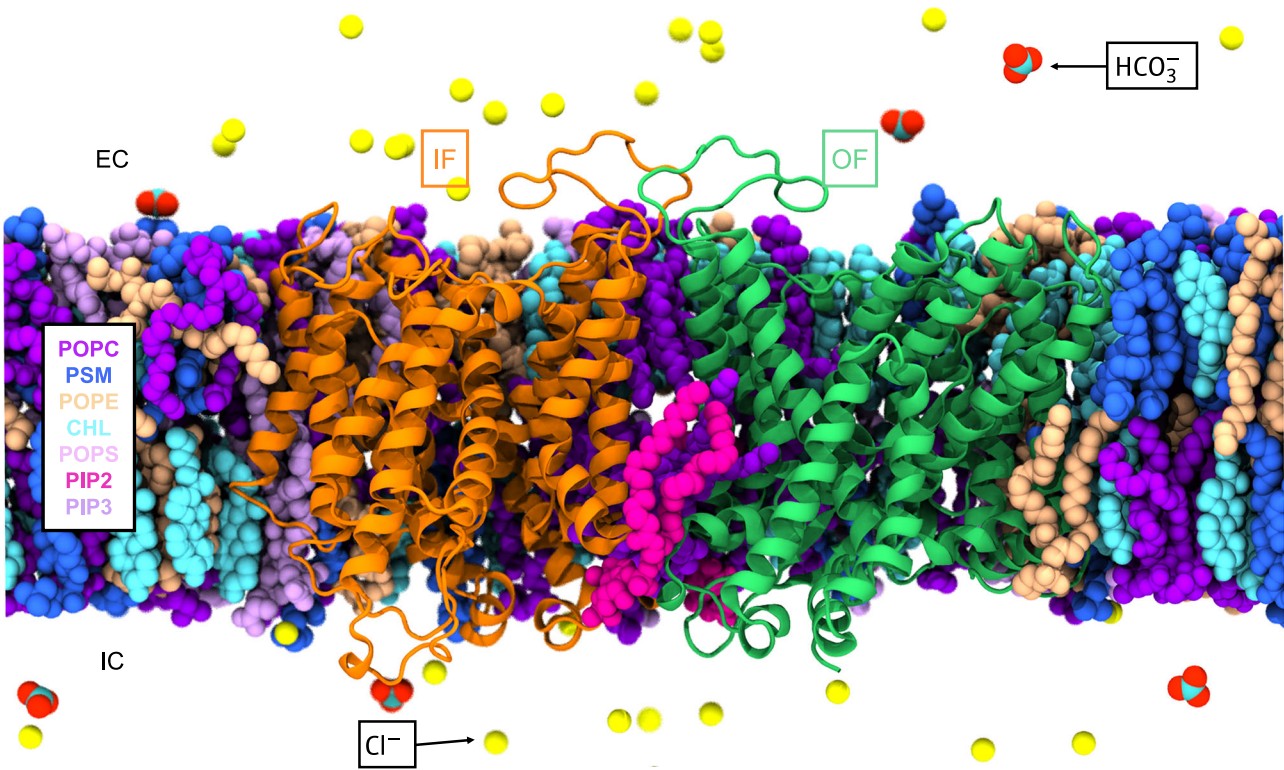

**Fig. 7 | Simulation system of membrane-embedded AE1 dimer.** The AE1 dimer embedded in a heterogeneous lipid membrane. Different lipid types included in the bilayer are represented in distinct colors as labeled, with a composition of cholesterol: phosphatidylcholine (PC): phosphatidylethanolamine (PE): sphingomyelin (SM): phosphatidylserine (PS): $PIP_2$: $PIP_3$ = 45:16:15:14:8:1:1. The two AE1 protomers are shown in cartoon representations, with the OF one in green, and the IF1 in orange. The $Cl^-$ and $HCO_3^-$ ions are shown as spheres. The interfacial $PIP_2$ lipid is highlighted in pink.

indicated in the figure legends. The reactions were quenched by the addition of ice-cold 100 mM KPi, pH 6.6 and 100 mM LiCl and filtered through 0.45 μm nitrocellulose filters (Millipore). The dried filters were incubated in a scintillation cocktail, and the retained radioactivity on the filters was determined by scintillation counting (Hidex, SL300). The specific uptake activity of AE1 was determined by subtracting the background signal determined in control liposomes (lacking AE1) from the signal measured in AE1-containing proteoliposomes and normalized to the specific signal in the absence of the compound. AE1 was incubated for 1 h with 15 mg/mL poly-lysine (Millipore Sigma), 0.3 U/mL PLC (Invitrogen) before being incorporated into the proteoliposomes. 50 μM $PIP_2$ (Millipore Sigma) was added to the protoliposomes and incubated for 1 h, after protein incorporation. Then the uptake of 10 μM $NaH[^{14}C]O_3$ or $Na[^{125}I]$ by AE1-containing proteoliposomes was measured for 1 min. Statistical analysis of the data was performed in GraphPad Prism 10 as appropriate. Data are mean ± SEM of $n = 3$. $P$-values were obtained with a two-tailed Mann-Whitney test.

### Simulation system preparation

For MD simulations, we started with the IF1/OF and IF1/IF2 AE1 dimer structures from the cryo-EM experiments. The cryo-EM structure lacked coordinates for specific loops, which were constructed using the Rosetta ab-initio fragment assembly within the Robetta protein structure prediction server[58]. The protonation states of the titratable residues were determined with PROPKA3.1[59]. From the cryo-EM density map, several lipids, including $PIP_2$ and POPC, were discernible, either at the dimer interface or elsewhere near the protein. Only the coordinates for the lipid head groups and the first few carbons of their tails could be assigned based on the density map. The complete lipid tails were then reconstructed using PSFGEN plugin in VMD1.9.4[60] according to their topology. To represent the cellular environment, we generated a heterogeneous lipid bilayer using the CHARMM-GUI Membrane

Builder tool[61,62], designed to replicate the lipid composition in human RBCs with a total of 757 lipids (cholesterol: PC: phosphatidylethanolamine (PE): sphingomyelin (SM): phosphatidylserine (PS): $PIP_2$: $PIP_3$ = 45:16:15:14:8:1:1)[63,64]. The PPM (Positioning of Proteins in Membrane) Server was used to determine the initial orientation and insertion depth of AE1 in the membrane[65]. Upon combining the membrane and protein, we removed all lipids placed by CHARMM-GUI within 1 Å of the protein or resolved lipids. The experimentally resolved PCs and $PIP_2$s were preserved. Subsequently, the membrane-embedded protein systems were solvated with a solution including 105 mM $Cl^-$ using the VMD Autoionize plugin. 18 mM $Cl^-$ was substituted by $HCO_3^-$ subsequently with VMD PSFGEN plugin. $K^+$ ions were added for charge neutralization. The assembled systems spanned dimensions of approximately 145 × 145 × 125 Å$^3$, containing 276 k atoms. The system setup information is in Supplementary Table S2. Lastly, to introduce variability in the lipids surrounding the protein in different simulation replicas, we generated multiple copies by shuffling all the lipids (excluding those from the cryo-EM), employing the VMD Membrane Mixer plugin[66] (Fig. 7).

### Equilibrium simulations

MD simulations were performed using a highly scalable MD engine, NAMD3.0[67,68], using the CHARMM36 and CHARMM36m forcefield parameters[69,70]. The usage of all-atom, fixed-charge force fields, in combination with explicit TIP3P water model has been extensively validated in previous studies for simulating membrane proteins[39,40]. Simulations were conducted under the NPT ensemble, maintaining the temperature at 310 K with the Langevin thermostat[71]. The Nose–Hoover Langevin piston method was used to keep the pressure at 1 bar[72,73]. Non-bonded interactions were computed with a 12 Å cutoff and a switching distance starting at 10 Å. Long-range electrostatic interactions were computed using the particle mesh Ewald method

under periodic boundary conditions[74,75]. Bonds involving hydrogen atoms were fixed using SHAKE and SETTLE algorithms[76]. A 2-fs timestep was utilized for both equilibrium simulations and production runs, unless otherwise specified.

After system preparation, we initiated a five-phase equilibration. During the initial phase, harmonic restraints of 10 kcal/mol/Å² were applied to protein backbone heavy atoms, and 5 kcal/mol/Å² to protein side chain and lipid head-group heavy atoms. This stage included 10,000 steps of conjugate gradient energy minimization followed by a 100 ps MD simulation with a 0.5-fs timestep. For the subsequent three phases, which were each conducted for 500 ps, we employed a 1-fs timestep with sequentially halving the restraints on the protein and lipids. In the final stages, the systems were equilibrated for 1 ns with a 2-fs timestep. Upon completing the equilibration stages, the production runs were performed, with all restraints removed, for eight replicas of IF1/OF and IF1/IF2 systems, each spanning 1 μs.

### Characterization of anion binding

Anions were considered interacting with AE1 when they were within 3.5 Å of any heavy atom of the protein. Anion density maps were generated by analyzing the averaged presence of anions in all 16 simulations using the volumetric map tool in VMD. The ranking of the luminal anion-interacting residues was based on the fraction of frames in which anions bound to specific luminal residues. The dissociation constant $K_d$ was estimated using the equation $K_d = t_{unbound}^{total} \cdot [HCO_3^-]/t_{bound}^{total}$, which is derived from the equation $K_d = k_{off}/k_{on}$, where $k_{off} = 1/\tau_{bound}$ and $k_{on} = 1/(\tau_{unbound} \cdot [HCO_3^-])$, and $\tau$ values represent the average times spent in the bound or unbound states.

### Simulating the transition with biasing simulations

The conformational transition from the IF1/OF structure to the IF1/IF2 one was induced using driven (biased) simulations. In this approach, biases are strategically applied to select atoms, promoting specific structural alterations of the system. We employed two system-specific collective variables (CVs)[38] through the CV module in NAMD[67,68]. CVs describe various geometric relationships ranging from the basic ones, such as center of mass (COM) distances between two atom groups, to more complex ones, such as radius of gyration or relative orientations of different structural elements. The selection of CVs was driven by empirical considerations and a comparative analysis of the OF and IF structures. The translational and rotational motions of the TD relative to the SD can nearly completely describe the conformational transition of AE1 between the IF and OF states. These motions can be best quantified by CVs $z$ and $\theta$[77], as described in Fig. 4A. $z$ represents the distance between the COMs of the TD in the OF and the IF2 states when the systems are aligned using the SD. $\theta$ is the angle of the first principal axis of the TM helices of the TD relative to that in the IF structure when aligned with SD. The calculations of COMs, angles, and alignments (using the SDs) involve only the Cα atoms of the TM helices.

Three systems were set up: *apo*, $HCO_3^-$-bound, and $HCO_3^-$-bound with cryo-EM $PIP_2$ lipids removed. The biasing restraints and pulling protocols were the same for all three systems. In the $HCO_3^-$-bound system, an extra bond was introduced to effectively anchor the $HCO_3^-$ to the pivotal binding residue, R730. This was done by a harmonic restraint ($k = 0.2$ kcal/mol/Å²) between the carbon atom of $HCO_3^-$ and the NH1 atom of R730 side chain, preventing the substrate from unbinding and diffusing into the solution during the transition. For each system, we first performed a 500 ns equilibrium simulation. Then a driven simulation was conducted for 200 ns, followed by a 100-ns releasing stage in which the force constants of the two CVs were gradually reduced. The final structures were then equilibrated for another 500 ns without any biases to ensure their stability.

### Refinement of the transition pathways using string method with swarms of trajectories (SMwST)

The transition pathways obtained from the driven simulations are inherently biased by the CV protocol used, and need to be relaxed to capture more energetically favorable transitions. SMwST functions as a path-refinement algorithm optimized for multidimensional CVs[42,78,79]. It employs a multi-walker simulation, where each walker iteratively samples and fine-tunes a point in the initial transition pathway. The process is iterated until convergence, which is represented by the stabilization of a local minimum-energy pathway within the defined CV space. With SMwST, the initial pathway is segmented into N equidistant points termed as images. The positions of these images are refined by averaging over the net drift across M identical simulation replicas performed on them, named as the swarms. The AE1 transition pathways were each divided into 32 images (or windows), and 24 runs were launched from each image to sample regions near the trajectory in the CV space. Every cycle encompassed two stages: a 10-ps restrained simulation that anchored the M replicas to the current center, followed by a 10-ps unrestrained MD to allow for drift. These two-stage cycles facilitated drifting along the local free energy gradient. After each cycle, updates are made to the image centers, complemented by a smoothing and reparametrization procedure. For the AE1 transitions, we continued the SMwST runs for a total of 200 cycles on each system, although the convergence was achieved after around 100 iterations in all of them.

### Calculating the free energy of the transition with bias-exchange umbrella sampling (BEUS)

To determine the free energy profile, or the potential of mean force (PMF), along the refined pathway, we employed 1D-BEUS calculations[41,43]. The refined trajectory obtained from SMwST was segmented into 64 windows (replicas). Each window was then simulated with a bias to its current center in the CV space, applied using a harmonic restraint with a force constant of 36 kcal/mol/Å². During the simulation, replica exchange was attempted every 1 ns, allowing for the swapping of the restraining centers between replicas in adjacent windows based on a Monte Carlo algorithm, thereby enhancing sampling efficiency. Each replica was sampled for 20 ns, resulting in a cumulative simulation time of 1.28 μs. The convergence of the simulations and sufficient overlap between adjacent windows were confirmed by examining the histograms along each CV (Supplementary Fig. S8D, E). The sampled data were reweighted using the Weighted Histogram Analysis Method (WHAM) to generate the free energy profiles[80].

### Ethical Statement

Ethical approval was obtained for the study from the University of Padua. Red blood cells were obtained from expired whole blood donations to the Transfusion center of the University Hospital of Padua. Donors were informed prior to donation that anonymized, expired donations could be used for research purposes. No additional samples were collected for research purposes. Red blood cells were received and processed to generate ghost membranes by the Calì lab at the University of Padua (Dept. of Biomedical Sciences).

### Reporting summary

Further information on research design is available in the Nature Portfolio Reporting Summary linked to this article.

## Data availability

The cryo-EM maps have been deposited in the Electron Microscopy Data Bank (EMDB) under accession codes: EMD-48421 (AE1 IF1-OF). EMD-48422 (AE1 OF-OF). EMD-48480 (AE1 IF1-IF2). The atomic coordinates have been deposited in the Protein Data Bank (PDB) under accession codes: PDB 9MND (AE1 IF1-OF). PDB 9MNG (AE1 OF-OF). PDB 9MOS (AE1 IF1-IF2). The atomic model of AE1 used for model

building is available as PDB 4YZF. The simulation input files, structure files, parameters, and truncated MD trajectories have been deposited in an open public repository [https://doi.org/10.5281/zenodo.13863897]. The source data underlying Figures and Supplementary Fig. are provided as a SourceData.zip file deposited in the open public repository [https://doi.org/10.5281/zenodo.13863897] and is available with this manuscript. Source data are provided with this paper.

## Code availability

All simulation input scripts, analysis tools, and any custom code developed in this study are available at an open public repository [https://doi.org/10.5281/zenodo.13863897].

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

## Acknowledgements

This study was supported by the National Institutes of Health (NIH) through grants P41-GM104601 (E.T.), R24GM145965 (E.T.), R01-HL168178 (E.T. and O.B.C.), and R01-DK128315 (E.T.). Support for O.B.C. was provided by the Irma T. Hirschl Trust. Computational simulations were conducted using resources provided by the National Science Foundation Supercomputing Centers (ACCESS grant number MCA06N060), and Delta advanced computing and data resource which is supported by the National Science Foundation (award OAC 2005572) and the State of Illinois. We acknowledge the Columbia Cryo-EM Center, particularly Robert Grassucci and Zhening Zhang, for their assistance with data collection.

## Author contributions

T.C., E.T, F.V., and O.B.C. conceived the study. T.C., E.T. F.V., E.G., M.Q., and O.B.C. designed the experiments. T.C. and E.T. performed the molecular dynamics simulations. T. Calì prepared ghost membranes and provided input during manuscript preparation. F.V. and O.B.C. performed structural biology experiments, F.V., K.K., and O.B.C. built and refined structural models. E.G. and M.Q. performed biochemistry experiments. All authors analyzed the results. T.C. and E.T. wrote the manuscript with input from all authors.

## Competing interests

The authors declare no competing interests.
