## [Transparent Peer Review file · Nature Communications]

Impact of anionic lipids on the energy landscape of conformational transition in anion exchanger 1 (AE1)

Corresponding Author: Professor Emad Tajkhorshid

Version 0:

Reviewer comments:

Reviewer #1

(Remarks to the Author)

This paper describes the structural changes involved in the membrane transport of anion exchanger 1 (AE1) in red blood cell (RBC) membranes. The protein works for the counter-transport of HCO₃⁻ and Cl⁻. In this study, they reveal two structures of the inward-facing (IF) and one outward-facing (OF) state by structural analysis of the transmembrane region of AE1 using cryo-EM. They also executed an anion uptake assay in the presence of phosphatidylinositol 4,5-bisphosphate (PIP2) lipids. Furthermore, molecular dynamics (MD) calculations and path sampling simulations discuss the role of substrate (HCO₃⁻) and PIP2 in the structural changes of IF/OF. This is an interesting study with a new approach, but some points need to be revised, and some points need to be described in more detail. These points are described below.

1. From the main text, it seems that the structure of AE1 obtained by cryo-EM is only the TMD region. If this is the case, the figure caption for Figure 1 should clearly state this. Furthermore, this point is not clearly stated in the ion uptake assay or simulations and should be explained.
2. In Figures 1, 2C, 3A, S2, and the supplemental movie, please clearly indicate which side is the cytoplasmic part and which side is outside the cell.
3. Please also add an explanation of the supplemental movie. Was the transport of HCO₃⁻ and the dissociation of hydrogen bonds between the PIP2 and K743 observed in a single MD simulation? If so, please write which structure (OF/IF) you observed. In the movie, HCO₃⁻ seems to be transported into the cell from the outside, but physiologically, I think that HCO₃⁻ can be transported from the cell to the outside, too. If so, was the export observed?
4. Please describe why the simulation starts with IF2/OF (Page 16, second paragraph, first line) as the initial structure, even though one of the dimer structures obtained by Cryo-EM is IF1/OF, and IF1 is the predominant structure. Also, the methods should describe how the dimer structure was created.
5. Please describe the equation used to estimate dissociation constant K_d. Did you calculate the rate of dissociation and binding for the substrate? Also, please discuss the reasons for the large discrepancy with the NMR experimental values.
6. What is the reason for the large difference in the binding probabilities of Cl⁻ and HCO₃⁻?
7. Although it is written as if the CV, z is equal to the distance of com, but in fact it is not the distance, as can be seen in Fig. 4A. In Figure 4D and other figures, you wrote z as 'Translation' or 'Transport domain translation'. Please describe the CV correctly.
8. Please show the overall movement (rotation angle and translation) of important helices such as TM10 and TM3 on the pathway, as well as changes in CV and the distance between residues, about the structural changes in IF/OF. Please discuss whether the presence or absence of substrate (HCO₃⁻) and PIP2 affects the overall movement.
9. I don't understand what Figure S7 is supposed to mean, so please consider the revision of the graph (tics of x-axis), layout of graphs, and captions. Also, the part of the text that explains the graph (in the 1st paragraph on page 12) is unclear.

The following are minor revisions.

1. In the first paragraph on page 15, please enter the correct IDs for EMDB and EMPIAR.
2. In Table S1, please enter the correct IDs for PDB. IF and IFINT should be IF1 and IF2 in the table.
3. In Figure 3, the two TMs are written as a-3 and a-10, but are they the same as TM3 and TM10? If so, please unify them. Please write that a green stick represents Glycan.
4. The negative sign of the z-coordinate in Figure 3A has disappeared. Also, the Z-coordinates here are written as z, and it is

difficult to understand because z, which is one of the CVs, is written in the same way.

5. In Figures 4C, D, E, S3, and S4, you use “rotation”, “translation”, “transport domain rotation”, and “transport domain translation”, the same as CV’s z and θ ? If so, please unify them.

6. In Figures 5A and C, please clarify whether the color bars in A and C are 0-31 or 0-32. Please clearly state in the title which window “the transition state” in Figures 5B and D is from.

7. In Figure 6, the captions for B and D seem to describe the graphs for E and F. Please correct them.

Reviewer #2

(Remarks to the Author)

Please refer to the attachment with highlighted area.

The manuscript by Chen and colleagues reported three distinct hAE1 TMD conformations: IF1, IF2 and OF. In IF1 the TM10 helix is unfolded, while in IF2 TM10 is fully ordered. In the OF TM10 is also well-ordered. The inhibitory role of PIP2 on ion transport was investigated. Further, MD simulations were performed to analyze the ion binding effect during conformational transition. However, there are many significant discrepancies that needs to be addressed as outlined in my comments below.

Major

1. “During this mechanism, the movements of the gate and core domains carry the substrate from one side of the membrane to the other”.

In elevator type movement, the core domain usually undergoes positional displacement, while the gate domain keeps relatively immobile, rather than both gate and core domains move.

2. “while in IF2, TM10 is fully ordered, with R730 interacting with the C-terminal end of the TM3 helix (Fig. 1C).”

It not the C-terminal end but the N-terminal end of TM3. The authors should correct this by looking into the structure carefully.

3. “Multiple different configurations of the AE1 dimer were separated by 3D classification, of which three major configurations were refined to high resolution: IF/OF at 2.4 Å, OF/OF at 3.1 Å, and IF1/IF2 configuration at 2.9 Å resolution.”

It should be IF1/OF based on Figure S9. The authors should address it properly to keep the consistency throughout the text.

4. “AE1 monomers in the IF1 state represent 48%, the OF conformation accounts for 32%, and the IF2 state represents 20% of the total particles.”

The authors should address the percentage carefully by calculating correct numbers. Refer to both Figure S9 and Supplementary Table, IF1-OF dimer consists of 666k particles, 109k of OF-OF dimer, 79k of IF1-IF2 dimer. So, the number of IF1 monomer is 745k (666k+79k), accounting for 43.6% of all monomer particles. 51.8% for OF monomer and 4.6% for IF2 monomer.

5. “which is alternately accessible to either side of the membrane in the OF and IF states through the elevator movement of the transporter’s gate and core domain . 25 The gate domain undergoes the most notable change, particularly in the region that involves TM10.”

As is known, in the IF bAE1 (ref16) and IF AE2 (ref22), the core domain undergoes the most notable change during conformational transition while the gate domain keeps relatively immobile.

6. “Recent studies have demonstrated the involvement of this residue in coordinating HCO – 3 within the binding pocket of AE1.”

Lack of proper reference.

7. “We also observed a second IF state, IF2, in which R730 is directed towards the intracellular space but displaced by only 7 Å compared to the OF states.”

It does not look like that R730 is directed towards the intracellular space to me by looking at the middle image of Figure 1C. Please clarify it.

8. “A comparison of the OF and IF conformations reveals that the transition between the states occurs via rotation of the gate domain around an axis roughly intersecting with the intracellular distal corner (F507).”

Again, it is not the gate domain. Also, the authors stated that “A comparison of the OF and IF conformations” but I did not see the comparison in any of the figures and supporting information. The authors should address this important information in Figure 1. Further, the description on OF and IF transition is not sufficient and unclear.

9. Figure 1. There are several concerns that need to be corrected carefully. 1) About the colors used for two monomers. Dark and light green are too similar to each other. 2) Panel A and B should be rearranged. Cryo-EM maps should go first, followed by models. 3) figure legend “The IF1/OF AE1 dimer viewed from the membrane plane (left) and from the cytoplasm (right).” It is actually not the cytoplasmic side. The left image shows the view from the extracellular side. 4) The 90 ° rotation symbol is in the wrong direction, which is not consistent with the left image. 5) Missing key labels. Regardless of every single TM, the authors should at least label the core and gate domain. 6) in panel C, the left schematic representation of IF1, the authors still presented TM10 as α helix, which is misleading.

10. “Lipid headgroup densities are seen at the intracellular dimer interface on the two sides of the protein, where they bind to patches of positively charged residues 2.”

I do not understand what is residues 2. Please clarify.

11. “Several amino acids involved in PIP2 binding are conserved within the SLC4 transporter family, particularly P815 and positively charged residues R602 and K817”

The authors demonstrated that R602 is crucial to PIP2 binding, but they did not show R602 in the Figure 2A and B. Please clarify.

12. “When mutated to histidine, it causes recessive distal renal tubular acidosis. 29 This suggests that the positive charge at position 602 may be essential for stabilizing the AE1-PIP2 interaction and, consequently, the dimer interface.”

More evidence is needed to support the statement. Shown in Figure B, K600 is closer to PIP2 than R602. Except for R603,

both K600 and R603 are positively charged. The dimeric interaction is mostly contributed by hydrophobic interactions of TM6s from two monomers as interpreted in published AE1, AE2, AE3, NBCe1, NDCBE structures.

13. Figure 2 legend error. "In (D), 50 μ M PIP2 was added to liposomes previously treated with PLC."

Which should be In (F)

14. "To test whether PIP2 affects the stability of AE1, we first checked its possible effects on substrate transport using a proteoliposome-based assay. We reconstituted AE1 into liposomes and measured the uptake of NaHCO₃ and NaI to assess the transport activity for the two of the established AE1 substrates. 30 To determine AE1's affinity for HCO₃⁻ and I⁻, isotopic dilutions of the radiolabeled anion with increasing concentrations of the unlabeled anion were performed (Supplementary Fig. S1A), yielding an EC₅₀ (effective concentration of the unlabeled anion to reduce the uptake of radiolabeled anion by 50%) of 79 nM for HCO₃⁻ and 143 nM for I⁻. We verified AE1-specific uptake by performing the assays in the presence of DIDS (4,4'-Diisothiocyano-2,2'-stilbenedisulfonic acid), a known anion transporter inhibitor. 31 The uptake levels in the presence of DIDS were reduced by half compared to those without the inhibitor (Supplementary Fig. S1B)."

What is the conclusion of the effect of PIP2 on AE1 stability? Is the assay carried out here relevant to AE1's stability?

15. How the authors control the orientation of the transporter in liposome, i.e., inside-in or inside-out? If they can not, how do they determine the inside-in/inside-out ratio?

16. "To gain insight into the binding of anions within the lumen formed between the scaffold domain (SD) and transport domain (TD)"

The authors first used gate and core domain, now turned into SD and TD. Should be correct to keep consistency.

17. The estimated dissociation constants K_d for Cl⁻ and HCO₃⁻ (255mM and 22mM for OF, 364mM and 38 for IF) are quite high. Do these values reflect the real physiological conditions or may they be influenced by the specific system used in the simulations?

18. The statements of "AE1 monomers in the IF1 state represent 48%, the OF conformation accounts for 32%, and the IF2 state represents 20% of the total particles." and "The elevated free energy of the IF state could explain its lower prevalence in our and previous structural studies. 17–19" are inconsistent. The authors stated first IF1 state represent 48%, OF 32% and IF2 20%, then stated that IF has lower prevalence in their and previous structural studies. Please correct according to my comment 5.

19. Incomplete Figure 6 legend. Missing E and F figure legend. Figure 6B legend is incorrect.

20. The authors stated that substrate binding can lower the energy barrier from OF to IF and the salt bridge between K743 and the 4- or 5-phosphate groups in PIP2 needs to break before reaching the IF state. In the method section, the AE1 sample is prepared without HCO₃⁻ and without PIP2 removal. But the authors can still get IF1 and IF2 by adding 2.5 mM DSP (dithiobis(succinimidyl propionate)) crosslinker. My question is how can the DSP increasing the prevalence of IF states, where there is no HCO₃⁻ which means high energy barrier between OF-IF, and salt-bridge between PIP2 and K743 present?

Minor

1. Data availability is not accessible. "Maps and raw movies have been deposited at EMDB (IDs: xxx) and EMPIAR (IDs: yyy)."

2. Clash score is too high for corresponding resolutions provided.

3. EMRinger score should be provided for model validation.

4. Numerous grammar errors need to be carefully corrected.

5. Some format issues. For example: "Membranes were washed 3 times with low-salt buffer (10 mM HEPES, 0.1 M KCl, 0.5 mM dithiothreitol (DTT))" and "The sample was then loaded on a glycerol gradient (12-28% glycerol in HEPES 10 mM, NaCl 130 mM and 0.05% Triton X-100, pH 7.5)"

6. Figure S12, the authors should provide the model/map fit for every TM (1-14)

Version 1:

Reviewer comments:

Reviewer #1

(Remarks to the Author)

The revised manuscript has been improved significantly.

We don't have any concerns about the revised manuscript. It is a nice simulation study on the anion exchanger 1.

Reviewer #2

(Remarks to the Author)

Thank you for addressing my concerns and correcting errors that mentioned.

In the manuscript, three conformations were reported, i.e. IF1-OF, OF-OF and IF1-IF2. The structures were resolved by DSP cross-linking, which absolutely changed the IF prevalence. However, bicarbonate --one of the substrate was not added in the sample, the authors are hypothesizing that AE1 undergoes OF to IF transition over time without bicarbonate according to answer for comment 20. If so, AE1 can transport chloride only, which is not the truth. Something behind remains unknown.

In the paper titled Cryo-EM structures of the human band 3 transporter indicate a transport mechanism involving the coupled

movement of chloride and bicarbonate ions (ref 20), published on Plos Biology Aug 21, 2024. It reported all the conformations you have and more, i.e. IF-OF-Cl-, IF-IF-Cl-, IF-OF-HCO₃-, IF-IF-HCO₃-, OF-OF-HCO₃-. In addition, all the reported structures were resolved in their native states without cross-linking. What kind of innovation would make this work publishable in Nature Communications?

Version 2:

Reviewer comments:

Reviewer #2

(Remarks to the Author)

Thank you for the authors' feedback. It addresses my concerns properly. The only issue I found is that the cryoEM map of IF-OF provided by the author has very poor density in the region of TM10, which is disordered in IF1. It is unlikely to track the main chain of TM10 based on the cryoEM map. I am wondering how did the author build the atomic model of this region and determine the orientation of R730.

Version 3:

Reviewer comments:

Reviewer #2

(Remarks to the Author)

Thank you for the authors' answer.

Response to the Reviewers

Reviewer 1

Reviewer: This paper describes the structural changes involved in the membrane transport of anion exchanger 1 (AE1) in red blood cell (RBC) membranes. The protein works for the counter-transport of HCO_3^- and Cl^- . In this study, they reveal two structures of the inward-facing (IF) and one outward-facing (OF) state by structural analysis of the transmembrane region of AE1 using cryo-EM. They also executed an anion uptake assay in the presence of phosphatidylinositol 4,5-bisphosphate (PIP_2) lipids. Furthermore, molecular dynamics (MD) calculations and path sampling simulations discuss the role of substrate (HCO_3^-) and PIP_2 in the structural changes of IF/OF. This is an interesting study with a new approach, but some points need to be revised and some points need to be described in more detail. These points are described below.

Response: We thank the reviewers for nicely summarizing the results reported in our manuscript.

Reviewer: 1. From the main text, it seems that the structure of AE1 obtained by cryo-EM is only the TMD region. If this is the case, the figure caption for Figure 1 should clearly state this. Furthermore, this point is not clearly stated in the ion uptake assay or simulations and should be explained.

Response: The AE1 we used was purified from human erythrocytes and contains both the cytoplasmic and transmembrane domains. However, the cytoplasmic domain is highly mobile and can generally only be resolved at high resolution when bound to other proteins (e.g. in complex with ankyrin-1). The uptake assay was performed with full-length AE1 and MD simulations were conducted with the TMD region only. The statement is added to the Figure 1 caption and we have added the following to the main text to clarify this point:

“We purified the full-length AE1 dimer from human erythrocytes, including both the CD and the TMD. However, we were only able to resolve a high-resolution structure for the TMD part of the protein, as the CD is known to be extremely mobile (Supplementary Fig. S1A). High-resolution visualization of CD is typically possible only when it binds to other proteins.¹⁰”

“We reconstituted full-length AE1 into liposomes ...”

“we conducted 8 independent 1- μs simulations of the TMD of ...”

Reviewer: 2. In Figures 1, 2C, 3A, S2, and the supplemental movie, please clearly indicate which side is the cytoplasmic part and which side is outside the cell.

Response: We thank the reviewer for pointing out this important point. We have added this information to the revised Figs. 1, 2C, 3A, and S3 (formerly S2), and the supplemental movie.

Reviewer: 3. Please also add an explanation of the supplemental movie. Was the transport of HCO_3^- and the dissociation of hydrogen bonds between the PIP_2 and K743 observed in a single MD simulation? If so, please write which structure (OF/IF) you observed. In the movie, HCO_3^- seems to be transported into the cell from the outside, but physiologically, I think that HCO_3^- can be transported from the cell to the outside, too. If so, was the export observed?

Response: The explanation is now added to the caption of the Supplementary Movie. The movie does not show the trajectory of a single equilibrium simulation. HCO_3^- transport and the dissociation of hydrogen bonds occurred during multiple stages of SMD simulations which induced the OF to IF2 transition as indicated in Fig. 4B. The camera in the movie focuses on the protomer that was initially in the OF state and underwent the transition. For the last question, due to the reversibility of the function of the transporter, the observed process can be also considered the reverse of the export of HCO_3^- from the cell to the outside (microscopic reversibility). Furthermore, we have performed an additional SMD simulation inducing the transition from IF2 to OF, and the transport of HCO_3^- in the opposite direction was observed in this simulation, whose results are depicted in Fig. S6B. The revised text and caption of the movie reads:

“Additional SMD simulations driving the reverse transition from the IF2 to OF state demonstrate that the HCO_3^- transport in the opposite direction is also feasible (Supplementary Fig. S6B).”

“Supplementary Movie 1: **Conformational changes, HCO_3^- transport, and PIP_2 interactions during the OF-IF transition.** In this movie, an extracellular HCO_3^- initially diffuses and binds to R730 of the AE1 protomer in the OF state. The protein then undergoes the OF to IF transition under the applied biases. Concurrently, the salt bridge between PIP_2 and K743 is disrupted. Finally, HCO_3^- dissociates from the binding site and is translocated to the cytoplasmic side once its restraint to R730 is released.”

Reviewer: 4. Please describe why the simulation starts with IF2/OF (Page 16, second paragraph, first line) as the initial structure, even though one of the dimer structures obtained by Cryo-EM is IF1/OF, and IF1 is the predominant structure. Also, the methods should describe how the dimer structure was created.

Response: We thank the reviewer for highlighting this point. Upon review, we acknowledge that we misstated the systems used for simulation. Equilibrium simulations were performed on both IF1/OF and IF1/IF2 dimeric systems, but the subsequent analyses focused on the OF and IF2 protomers. The main structural difference between the IF1 and IF2 conformations is the secondary structure of TM10. In the uncoiled (IF1) state, the putative binding residues on this helix do not form a defined binding site, so we didn't include this protomer in further analysis. The SMD simulations reported in the manuscript were performed with the IF1/OF structure as the starting point, where the OF protomer was transitioned to the IF2 state during the simulations, i.e., IF1/OF to IF1/IF2, ignoring the IF1 protomer. Initially, we attempted to simulate the transition from OF to IF1 by introducing an additional CV to uncoil TM10 into a loop conformation. However, this transition turned out to be highly costly (much more than the one reported in the manuscript). Therefore, we chose IF2 as the target state. We have revised the text as follows to clarify this aspect:

Results: “we conducted 8 independent 1 μs simulations of the membrane-embedded TMD of AE1 using the IF1/OF and IF1/IF2 structures individually. ... In the IF1 state, TM10 is completely unfolded, and putative substrate-binding residues do not form a well-defined binding site. No bound ion was present in this site in a recent structure.²⁰ Consequently, the analysis focused exclusively on the OF and IF2 protomers.”

“Conformational transition from the OF to the IF2 state was realized using driven simulations employing two CVs.”

Methods: “For MD simulations, we used the IF1/OF and IF1/IF2 AE1 dimer structures from the cryo-EM experiments. ... The conformational transition from the IF1/OF structure to the IF1/IF2 one was induced using driven (biased) simulations.”

Reviewer: 5. Please describe the equation used to estimate dissociation constant K_d . Did you calculate the rate of dissociation and binding for the substrate? Also, please discuss the reasons for the large discrepancy with the NMR experimental values.

Response: Correct. The dissociation constant K_d was calculated using equation $K_d = k_{\text{off}}/k_{\text{on}}$, where $k_{\text{off}} = 1/\tau_{\text{bound}}$ and $k_{\text{on}} = 1/(\tau_{\text{unbound}} \cdot [\text{HCO}_3^-])$ are obtained from the rates of binding and unbinding. Here, τ values represent the average time spent in the bound or unbound states. As a simplified estimate, $K_d = t_{\text{unbound}}^{\text{total}} \cdot [\text{HCO}_3^-] / t_{\text{bound}}^{\text{total}}$. We have modified the Methods as:

“The dissociation constant K_d was estimated using the equation $K_d = t_{\text{unbound}}^{\text{total}} \cdot [\text{HCO}_3^-] / t_{\text{bound}}^{\text{total}}$. It is derived from the equation $K_d = k_{\text{off}}/k_{\text{on}}$, where $k_{\text{off}} = 1/\tau_{\text{bound}}$ and $k_{\text{on}} = 1/(\tau_{\text{unbound}} \cdot [\text{HCO}_3^-])$, and τ values represent the average times spent in the bound or unbound states.”

Response: Regarding the second part of the question, the main source of discrepancy is the limited sampling time, even though we performed the simulations for 1 μs each. The limited timescale is comparable to or even shorter than the typical residence time of the substrate, which introduces a significant bias in the calculations. For instance, HCO_3^- can diffuse to the binding site and remain bound for the entire remainder of the simulation (no unbinding). In such cases, the measured τ_{bound} may be underestimated, while τ_{unbound} could be overestimated, as it includes the diffusion time to the binding site. These limitations introduce errors in the calculated average values, potentially contributing to the observed discrepancies with the experimental values of K_d . We have added this explanation to the revised manuscript to clarify this aspect:

“This discrepancy likely arises from the limited simulation timescale and the resulting small sample size for the binding events, leading to overestimated unbinding times due to diffusion effects and underestimated binding times.”

Reviewer: 6. What is the reason for the large difference in the binding probabilities of Cl^- and HCO_3^- ?

Response: We acknowledge that the large difference in the binding probabilities is a potential limitation of this study. We suspect the force field issues (use of a nonpolarizable forcefield) may be the main reason behind the observed difference. For instance, previous work (e.g., Orabi et al. 2021, JCTC, doi: 10.1021/acs.jctc.1c00550) has demonstrated that the charmm36m forcefield systematically underestimates interactions between Cl^- and protein residues. As a result, the calculated binding affinity of Cl^- may be artificially reduced, leading to a larger apparent dissociation constant (K_d) for it compared to HCO_3^- .

Reviewer: 7. Although it is written as if the CV, z is equal to the distance of com, but in fact it is not the distance, as can be seen in Fig. 4A. In Figure 4D and other figures, you wrote z as ‘Translation’ or ‘Transport domain translation’. Please describe the CV correctly.

Response: Thank you for pointing this out. The schematic representation of CV z in Fig. 4A was indeed inaccurately describing the CV. We have revised the schematic to accurately reflect the CV, and the text in Methods has been revised. Additionally, the labels in Fig. 4E and other panels have been updated for consistency and are now uniformly described as “Translation (\AA)”.

“ z represents the distance between the COMs of the TD in OF and the TD in IF2 when the systems are aligned using the SD. ... The calculations of COMs, angles and alignments (using the SDs) involve only the $C\alpha$ atoms of the TM helices.”

Reviewer: 8. Please show the overall movement (rotation angle and translation) of important helices such as TM10 and TM3 on the pathway, as well as changes in CV and the distance between residues, about the structural changes in IF/OF. Please discuss whether the presence or absence of substrate (HCO_3^-) and PIP_2 affects the overall movement.

Response: We have added a new figure (Supplementary Fig. S2 that shows the rotation and displacement of TM3 and TM10 during the refined pathways for the transition. The figure also shows important residue distances. TM3 motion remains consistent among different bound systems (panel C). In contrast, TM10 displays distinct behaviors in *apo*, HCO_3^- -bound, and PIP_2 -deleted systems (panel D): the presence of the substrate induces an earlier translation with less rotation in TM10, compared with the *apo* system. The removal of PIP_2 likely reduces the restraints on helix rotation, allowing TM10 to rotate more extensively alongside its translation compared to the HCO_3^- -bound system.

In addition, changes in inter-residue distances have been discussed in Fig. 5, where the R730-F532 and R730-F464 distances highlight significantly different interactions associated with substrate binding.

The collective movement of the TD has been shown in Fig. 4E. Overall, the trajectories exhibit no significant differences in the general motion of the TD across the three conditions. However, the presence of HCO_3^- induces slight deviations in the mid-region of the movement profile, while the removal of PIP_2 results in a small reduction in the TD’s translational displacement, compared to the other two systems.

We have added the following discussions in the text:

“A comparison of the CV changes among the three systems reveals that the presence of HCO_3^- induces subtle deviations in the mid-region of the movement profile of the TD, while the absence of PIP_2 leads to a slight reduction in the TD’s translational displacement (Fig. 4E). These findings suggest that both HCO_3^- and PIP_2 exert modest influences on the transition profile of TD, although their overall impact remains minimal.”

“We also observed a second IF state, IF2, in which R730 is directed towards the internal cavity and inaccessible from the extracellular space, with a displacement of only 7 \AA compared to the OF state (Supplementary Fig. S2A). Meanwhile, TM3 maintains the same position in the IF1 and IF2 structures but shifts upward by 8 \AA and rotates by 19° in the OF state (Supplementary Fig. S2B).”

“The rotation and displacement of TM3 (residues 466 to 482) and TM10 (residues 728 to 739), which form the binding site, along the refined trajectories were calculated by averaging across replicas for each SMwST window. The movement of TM3 remains consistent across different systems (Supplementary Fig. S2C), while for TM10, the presence of the substrate induces an earlier translation with reduced rotation compared to the *apo* system (Supplementary Fig. S2D). ... The removal of PIP₂ likely reduces the confinement of TM10 rotation, allowing it to rotate more extensively alongside its translation compared to the HCO₃⁻-bound system, while still showing earlier translational movement than the *apo* system (Supplementary Fig. S2D).”

Reviewer: 9. I don't understand what Figure S7 is supposed to mean, so please consider the revision of the graph (tics of x-axis), layout of graphs, and captions. Also, the part of the text that explains the graph (in the 1st paragraph on page 12) is unclear.

Response: As suggested by the reviewer, we have revised both the figure and the corresponding text to better clarify its content. The revised Fig. S8 (formerly S7) provides a direct comparison of the time-dependent fluctuations in RMSD values and their standard deviations for three types of lipids. The figure is intended to demonstrate the differences in RMSD during equilibration, highlighting the larger stability of PIP₂ compared to the other lipids, which is in support of PIP₂'s involvement in structural and functional properties of AE1. We have modified the text as follows:

“To evaluate the stability of the cryo-EM resolved lipids during the equilibrium simulation, we analyzed the RMSD of three distinct lipid groups: peripheral POPC lipids, which are located on the surface of the protein, POPC lipids at the dimer interface, and PIP₂ lipids. Supplementary Fig. S8 presents the headgroup RMSD values of these lipids relative to their initial positions, along with the corresponding standard deviations. Among these groups, PIP₂ lipids exhibit the lowest RMSD values and minimal variability, indicating greater stability even compared to POPC lipids confined at the dimer interface. This suggests that PIP₂ lipids are more likely to play a structural or functional role in the AE1 system.”

Response: The corresponding caption is changed to:

“**RMSD and corresponding standard deviation for cryo-EM resolved lipids in MD.** (A) Headgroup RMSD during the 1 μs equilibrium simulation of three groups of lipids relative to their initial position are shown: 8 POPC lipids peripheral to AE1 (orange), 4 POPC lipids within the dimer interface (blue), and 2 PIP₂ lipids (pink). (B) Standard deviation of the RMSD values for the same group, indicating the variability over the simulation time.”

Minor revisions

Reviewer: 1. In the first paragraph on page 15, please enter the correct IDs for EMDB and EMPIAR.

Response: We have deposited the three structures in the PDB and EMD databases and have added their IDs wherever they were missing. The details are as follows:

AE1 IF1-OF: PDB 9MND; EMD-48421

AE1 OF-OF: PDB 9MNG; EMD-48422

AE1 IF1-IF2: PDB 9MOS; EMD-48480

Reviewer: 2. In Table S1, please enter the correct IDs for PDB. IF and IFINT should be IF1 and IF2 in the table.

Response: We thank the reviewer for catching this. We have added the deposition IDs and corrected the nomenclature.

Reviewer: 3. In Figure 1, the two TMs are written as a-3 and a-10, but are they the same as TM3 and TM10? If so, please unify them. Please write that a green stick represents Glycan.

Response: They are the same and are unified in the text, and the glycan description is added to the Fig. 1 caption:

“The sticks in green and red represent N-linked glycosylation at residue N642 of each subunit.”

Reviewer: 4. The negative sign of the z -coordinate in Figure 3A has disappeared. Also, the Z -coordinates here are written as z , and it is difficult to understand because z , which is one of the CVs, is written in the same way.

Response: The proper sign is added to the figure and “ z -axis” is changed to “ Z -coordinates” to avoid misleading.

Reviewer: 5. In Figures 4C, D, E, S3, and S4, you use “rotation”, “translation”, “transport domain rotation”, and “transport domain translation”, the same as CV’s z and θ ? If so, please unify them.

Response: The reviewer is correct. These refer to the same thing as the CV. All the labels are changed to “Rotation/Translation” to be consistent.

Reviewer: 6. In Figures 5A and C, please clarify whether the color bars in A and C are 0-31 or 0-32. Please clearly state in the title which window “the transition state” in Figures 5B and D is from.

Response: We thank the reviewer for pointing this out. The color bar should be 0-31 since there are 32 windows in the simulation. The color bar in Fig. 5 is modified. The window numbers are also added in Fig. 5B and D.

Reviewer: 7. In Figure 6, the captions for B and D seem to describe the graphs for E and F. Please correct them.

Response: We thank the reviewer for pointing this out. The Fig. 6 caption is modified as:

“(A) Molecular representation of the PIP_2 - β -hairpin interaction in the OF state (green), highlighting PIP_2 in pink and K743 on the loop in licorice representation. (B) Enlarged view showing the positioning of K743 and PIP_2 in the OF state. (C) Molecular representation of the system after the protein transitions to the IF state. (D) Enlarged view showing the positioning of K743 and PIP_2 in the IF state. (E) Shortest heavy atom distances between PIP_2 and K743 during the 1 μs equilibrium simulation for the OF (red) and IF (blue) states. (F) Time evolution of the PIP_2 -K743 distance during driven simulations the protein undergoes transition from the OF to IF state.”

Reviewer 2

Reviewer: The manuscript by Chen and colleagues reported three distinct hAE1 TMD conformations: IF1, IF2 and OF. In IF1 the TM10 helix is unfolded, while in IF2 TM10 is fully ordered. In the OF TM10 is also well-ordered. The inhibitory role of PIP_2 on ion transport was investigated. Further, MD simulations were performed to analyze the ion binding effect during conformational transition. However, there are many significant discrepancies that need to be addressed as outlined in my comments below.

Reviewer: 1. “During this mechanism, the movements of the gate and core domains carry the substrate from one side of the membrane to the other”. In elevator type movement, the core domain usually undergoes positional displacement, while the gate domain keeps relatively immobile, rather than both gate and core domains move.

Response: We thank the reviewer for pointing out that this statement is inaccurate and can cause confusion. We have revised the text accordingly:

“During this mechanism, the movements of the TD carry the substrate from one side of the membrane to the other.”

Reviewer: 2. “while in IF2, TM10 is fully ordered, with R730 interacting with the C-terminal end of the TM3 helix (Fig. 1C).” It not the C-terminal end but the N-terminal end of TM3. The authors should correct this by looking into the structure carefully.

Response: We thank the reviewer for pointing out this mistake. We have modified the text as follows:

“... while in IF2, TM10 is fully ordered, with R730 interacting with the N-terminal end of the TM3 helix (Fig. 1).”

Reviewer: 3. “Multiple different configurations of the AE1 dimer were separated by 3D classification, of which three major configurations were refined to high resolution: IF/OF at 2.4 Å, OF/OF at 3.1 Å, and IF1/IF2 configuration at 2.9 Å resolution.” It should be IF1/OF based on Figure S9. The authors should address it properly to keep the consistency throughout the text.

Response: We thank the reviewer for this observation and have modified the text to ensure consistency:

“Multiple different configurations of the AE1 dimer were separated by 3D classification, of which three major configurations were refined to high resolution: IF1/OF at 2.4 Å, OF/OF at 3.1 Å, and IF1/IF2 at 2.9 Å resolution.”

Reviewer: 4. “AE1 monomers in the IF1 state represent 48%, the OF conformation accounts for 32%, and the IF2 state represents 20% of the total particles.” The authors should address the percentage carefully by calculating correct numbers. Refer to both Figure S9 and Supplementary Table, IF1-OF dimer consists of 666k particles, 109k of OF-OF dimer, 79k of IF1-IF2 dimer. So, the number of IF1 monomer is 745k (666k+79k), accounting for 43.6% of all monomer particles. 51.8% for OF monomer and 4.6% for IF2 monomer.

Response: We agree with the reviewer that this information was incorrect and have revised the text as suggested:

“AE1 monomers in the IF1 state represent 43.6%, the OF conformation accounts for 51.8%, and the IF2 state represents 4.6% of the total particles.”

Reviewer: 5. “which is alternately accessible to either side of the membrane in the OF and IF states through the elevator movement of the transporter’s gate and core domain. The gate domain undergoes the most notable change, particularly in the region that involves TM10.” As is known, in the IF bAE1 (ref16) and IF AE2 (ref22), the core domain undergoes the most notable change during conformational transition while the gate domain keeps relatively immobile.

Response: We thank the reviewer for pointing out that we inadvertently mixed up the gate and core domains on multiple occasions, creating confusion. We apologize for this oversight and have corrected it in the revised text:

“... which is alternately accessible to either side of the membrane in the OF and IF states through the elevator movement of the transporter’s TD. The TD undergoes the most notable change, particularly in the region that involves TM10.”

Reviewer: 6. “Recent studies have demonstrated the involvement of this residue in coordinating HCO_3^- within the binding pocket of AE1.” Lack of proper reference.

Response: We have added the proper reference to this sentence.

“Recent studies have demonstrated the involvement of this residue in coordinating HCO_3^- within the binding pocket of AE1.^{16,20}”

Reviewer: 7. “We also observed a second IF state, IF2, in which R730 is directed towards the intracellular space but displaced by only 7 Å compared to the OF states.” It does not look like that R730 is directed towards the intracellular space to me by looking at the middle image of Figure 1C. Please clarify it.

Response: We have corrected this in the text and added more details about the movement of R730:

“We also observed a second IF state, IF2, in which R730 is directed towards the internal cavity and inaccessible from the extracellular space, with a displacement of only 7 Å compared to the OF state (Supplementary Fig. S2A). Meanwhile, TM3 maintains the same position in the IF1 and IF2 structures but shifts upward by 8 Å and rotates by 19° in the OF state (Supplementary Fig. S2B).”

Reviewer: 8. “A comparison of the OF and IF conformations reveals that the transition between the states occurs via rotation of the gate domain around an axis roughly intersecting with the intracellular distal corner (F507).” Again, it is not the gate domain. Also, the authors stated that “A comparison of the OF and IF conformations” but I did not see the comparison in any of the figures and supporting information. The authors should address this important information in Figure 1. Further, the description on OF and IF transition is not sufficient and unclear.

Response: We thank the reviewer for pointing out the issue with the gate/core domain. We have corrected it in the text. Additionally, we have included a new figure (Supplementary Fig. S1B) to compare the two conformations and added further details in the text to better describe the transition:

“A comparison of the OF and IF conformations reveals that the transition between them occurs via TD’s movement around an axis roughly intersecting with the intracellular distal corner (F507) (Supplementary Fig. S1B).”

“As expected from prior studies, AE1 forms a homodimer. Analysis of different states within the same AE1 dimer clearly demonstrates that the two subunits function independently. The transport of Cl^- and HCO_3^- occurs at the interface between the two domains. While SD seems stationary between the outward-facing (OF) and inward-facing (IF) states, significant motion is observed in TD, particularly in TM3 and TM10.”

“This elevator-like motion is also reported for related transporters, e.g., SLC4A2.²⁴ Extra densities that correspond to phosphatidylcholine (PC) lipids (used for the nanodisc reconstitution) were modeled in the structure (Fig. 1A and B). We modeled also two PIP_2 lipids at the interface of the two AE1 subunits (Figs. 1 and 2).”

Reviewer: 9. Figure 1. There are several concerns that need to be corrected carefully. 1) About the colors used for two monomers. Dark and light green are too similar to each other. 2) Panel A and B should be rearranged. Cryo-EM maps should go first, followed by models. 3) figure legend “The IF1/OF AE1 dimer viewed from the membrane plane (left) and from the cytoplasm (right).” It is actually not the cytoplasmic side. The left image shows the view from the extracellular side. 4) The 90° rotation symbol is in the wrong direction, which is not consistent with the left image. 5) Missing key labels. Regardless of every single TM, the authors should at least label the core and gate domain. 6) in panel C, the left schematic representation of IF1, the authors still presented TM10 as α helix, which is misleading.

Response: We thank the reviewer for the detailed instruction as to how improve Fig. 1. We have revised the figure as follows: 1) We changed the colors in Fig. 1 (and all other figures for consistency) using orange for the IF1, blue for IF2 and light green for OF. 2) We switched panels A and B. 3) We modified the legend. 4) We changed the direction for the 90° rotation symbol in Fig. 1A-B. 5) We labeled and mentioned the different color of the core (TD) and gate domain (SD) in the figure legend. 6) We modified the representation in Fig. 1 panel C.

Reviewer: 10. “Lipid headgroup densities are seen at the intracellular dimer interface on the two sides of the protein, where they bind to patches of positively charged residues 2.” I do not understand what is residues 2. Please clarify.

Response: The number “2” was left in the text by mistake. We have removed it from the text.

Reviewer: 11. “Several amino acids involved in PIP_2 binding are conserved within the SLC4 transporter family, particularly P815 and positively charged residues R602 and K817” The authors demonstrated that R602 is crucial to PIP_2 binding, but they did not show R602 in the Figure 2A and B. Please clarify.

Response: We thank the reviewer for pointing out this omission. We have made a new Figure 2B, in which we have also included R602.

Reviewer: 12. “When mutated to histidine, it causes recessive distal renal tubular acidosis. This suggests that the positive charge at position 602 may be essential for stabilizing the AE1- PIP_2 interaction and, consequently, the dimer interface.” More evidence is needed to support the statement. Shown in Figure B, K600 is closer to PIP_2 than R602. Except for R603, hydrophobic interactions of TM6s from two monomers as interpreted in published AE1, AE2, AE3, NBCe1, NDCBE structures.

Response: We thank the reviewer for this point. R602 is now included in Fig. 2B. Our hypothesis is not that the mutation of R602 disrupts the interaction, but rather that it affects the stability of PIP_2 binding and, consequently, the dimer interface, as supported by our MD simulation data. We have modified the sentence to make it clearer:

“This suggests that the positive charge at position 602 may be essential for stabilizing the AE1- PIP_2 interaction and, consequently, the stability of the dimer interface.”

Reviewer: 13. Figure 2 legend error. “In (D), 50 μ M PIP₂ was added to liposomes previously treated with PLC.” Which should be In (F)

Response: The Reviewer is correct. The revised legend for Fig. 2 was changed accordingly:

“In (F), 50 μ M PIP₂ was added to liposomes previously treated with PLC.”

Reviewer: 14. “To test whether PIP₂ affects the stability of AE1, we first checked its possible effects on substrate transport using a proteoliposome-based assay. We reconstituted AE1 into liposomes and measured the uptake of NaH[¹⁴C]O₃ and Na[¹²⁵I] to assess the transport activity for the two of the established AE1 substrates. To determine AE1’s affinity for HCO₃⁻ and I⁻, isotopic dilutions of the radiolabeled anion with increasing concentrations of the unlabeled anion were performed (Supplementary Fig. S1A), yielding an EC50 (effective concentration of the unlabeled anion to reduce the uptake of radiolabeled anion by 50%) of 79 nM for HCO₃⁻ and 143 nM for I⁻. We verified AE1-specific uptake by performing the assays in the presence of DIDS (4,4’-Diisothiocyano-2,2’-stilbenedisulfonicacid), a known anion transporter inhibitor. The uptake levels in the presence of DIDS were reduced by half compared to those without the inhibitor (Supplementary Fig. S1B).” What is the conclusion of the effect of PIP₂ on AE1 stability? Is the assay carried out here relevant to AE1’s stability?

Response: This is an excellent point raised by the Reviewer. The approach used to test the effect on PIP₂ focuses on the activity rather than the stability. In the revised manuscript, we changed the sentence to:

“To test whether PIP₂ affects the activity of AE1, we first ... ”

Reviewer: 15. How the authors control the orientation of the transporter in liposome, i.e., inside-in or inside-out? If they can not, how do they determine the inside-in/inside-out ratio?

Response: The methodology used for the reconstitution of membrane proteins into proteoliposomes does not allow for controlling the orientation of the target protein in the bilayer. Subsequent ‘sorting’ approaches could be used, and were successfully used by our group (Fitzgerald et al., 2019, Nature), according to which proteoliposomes that contain a target protein in a particular orientation can be eliminated from the pool through specific chelating techniques (such as the use of a His-tag attached to the target protein that can be used to bind the fraction of proteoliposomes in which the His-tag is exposed to the external milieu). Also, in the 2019 Nature paper we could determine the inside-in/inside-out ratio based on specific labeling techniques using site-specific mutagenesis and site-directed thiol labeling in conjunction with a bacterial native cysteine-free protein background (MhsT). In contrast, AE1 was purified from natural sources without engineered tags, thus precluding such approaches. Secondary active transporters, such as AE1, use the free energy released from the energetically downhill translocation of one ion/substrate (A) to drive the translocation of another (B), the polarity of which depends on the direction of the substrate concentration gradient. In so far, in our approach the proteoliposomes were loaded with NaSO₄, and upon addition of the external test substrate (HCO₃⁻ or I⁻), the AE1-mediated ion exchange was monitored by the accumulation of radiolabeled HCO₃⁻ or I⁻ within the proteoliposomes. We note that, whereas we cannot rule out mixed kinetics of forward and reverse transport reactions for the test substrates in the ensemble assay, the determination of the transport activities under different experimental conditions was subject to the same ensemble bias and thus internally reliable.

Reviewer: 16. “To gain insight into the binding of anions within the lumen formed between the scaffold domain (SD) and transport domain (TD)” The authors first used gate and core domain, now turned into SD and TD. Should be correct to keep consistency.

Response: We thank the reviewer for pointing this out. The terminologies are made consistent as SD and TD in the whole text.

Reviewer: 17. The estimated dissociation constants K_d for Cl⁻ and HCO₃⁻ (255mM and 22mM for OF, 364mM and 38 for IF) are quite high. Do these values reflect the real physiological conditions or may they be influenced by the specific system used in the simulations?

Response: As mentioned in our response to Reviewer 1 (Comments 5 and 6), the calculated K_d values are likely influenced by several factors inherent to the molecular dynamics simulation setup. These include limitations of the simulation timescale, the finite binding sample sizes, and the non-polarizable force field. Such factors can introduce significant discrepancies between the calculated K_d values and physiological measurements. We have clarified this limitation in the manuscript as follows:

“This discrepancy likely arises from the limited simulation timescale and the resulting small sample sizes of binding, leading to overestimated unbinding times due to diffusion effects and underestimated binding times due to restricted simulation duration.”

Reviewer: 18. The statements of “AE1 monomers in the IF1 state represent 48%, the OF conformation accounts for 32%, and the IF2 state represents 20% of the total particles.” and “The elevated free energy of the IF state could explain its lower prevalence in our and previous structural studies.” are inconsistent. The authors stated first IF1 state represent 48%, OF 32% and IF2 20%, then stated that IF has lower prevalence in their and previous structural studies. Please correct according to my comment 4.

Response: Thank you for pointing out this issue, which we did not clarify properly. The ratio of the OF state (51.8%) is slightly higher based on the recalculated percentage according to comment 4. Moreover, the DSP crosslinker used during incubation may permanently biasing the percentage towards IF state as further explained in comment 20. To address it more effectively, we have added the following section to the text:

“The elevated free energy of the IF state could explain its lower prevalence in previous structural studies.^{10,17–19} The similar amount of particles in the OF or IF states observed in our structure is likely due to the incubation of the ghost membrane with the DSP crosslinker, which facilitates the crosslinking of lysine residues (K539 and K851) when the transporter adopts an inward-facing conformation. Once crosslinked, this modification becomes irreversible.”

Reviewer: 19. Incomplete Figure 6 legend. Missing E and F figure legend. Figure 6B legend is incorrect.

Response: Thank you for catching the inconsistencies in the legend of Fig. 6. We have revised the missing or incorrect legends and the updated Fig. 6 legend now reads as follows:

“(A) Molecular representation of the PIP₂- β -hairpin interaction in the OF state (green), highlighting PIP₂ in pink and K743 on the loop in licorice representation. (B) Enlarged view showing the positioning of K743 and PIP₂ in the OF state. (C) Molecular representation of the system after the protein transitions to the IF state. (D) Enlarged view showing the positioning of K743 and PIP₂ in the IF state. (E) Shortest heavy atom distances between PIP₂ and K743 during the 1- μ s equilibrium simulation for the OF (red) and IF (blue) states. (F) Time evolution of the PIP₂-K743 distance when the protein transitions from the OF to IF state.”

Reviewer: 20. The authors stated that substrate binding can lower the energy barrier from OF to IF and the salt bridge between K743 and the 4- or 5-phosphate groups in PIP₂ needs to break before reaching the IF state. In the method section, the AE1 sample is prepared without HCO₃⁻ and without PIP₂ removal. But the authors can still get IF1 and IF2 by adding 2.5 mM DSP (dithiobis(succinimidyl propionate)) crosslinker. My question is how can the DSP increasing the prevalence of IF states, where there is no HCO₃⁻ which means high energy barrier between OF-IF, and salt-bridge between PIP₂ and K743 present?

Response: although the energy barrier is higher in the absence of the substrate, the transition can still take place. To answer the reviewer’s question, “How can DSP increase the prevalence of IF states,” we observe clear cross-linking when the protomer is in the IF state, where the two cross-linked lysine residues (K539 and K851) are 8 Å apart. In contrast, in the OF state, where the same lysine residues are 10.5 Å apart, we do not detect any cross-linking. Our hypothesis is that even though the prevalence of the IF state may be low, covalent crosslinking of solely the IF state leads to increased population of this state over time.

Minor revisions

Reviewer: 1. Data availability is not accessible. “Maps and raw movies have been deposited at EMDB (IDs: xxx) and EMPIAR (IDs: yyy).”

Response: We have deposited the three structures in the PDB and EMD databases and have added their IDs wherever they were missing. The details are as follows:

AE1 IF1-OF: PDB 9MND; EMD-48421

AE1 OF-OF: PDB 9MNG; EMD-48422

AE1 IF1-IF2: PDB 9MOS; EMD-48480

The movie is available from the journal.

Reviewer: 2. Clash score is too high for corresponding resolutions provided.

Response: We agree with the reviewer and thank them for noticing it. We have finalized our structures and updated the validation details. As shown in Table 1, the clash scores for the three structures have significantly improved:

AE1 IF1-OF: 2.77

AE1 OF-OF: 4.5

AE1 IF1-IF2: 5.53

Reviewer: 3. EMRinger score should be provided for model validation.

Response: We are grateful to the reviewer for this suggestion. We have added the EMRinger scores for all the models in Table 1:

AE1 IF1-OF: 3.89

AE1 OF-OF: 4.44

AE1 IF1-IF2: 3.55

Reviewer: 4. Numerous grammar errors need to be carefully corrected.

Response: We have re-read and revised the whole manuscript to address this issue.

Reviewer: 5. Some format issues. For example: “Membranes were washed 3 times with low-salt buffer (10 mM HEPES, 0.1 M KCl, 0.5 mM dithiothreitol (DTT))” and “The sample was then loaded on a glycerol gradient (12-28% glycerol in HEPES 10 mM, NaCl 130 mM and 0.05% Triton X-100, pH 7.5)”

Response: We thank the reviewer for pointing it out. We have modified the text as follows:

“The sample was then loaded on a glycerol gradient (12-28% glycerol in 10 mM HEPES, 130 mM NaCl and 0.05% Triton X-100, pH 7.5)”

Reviewer: 6. Figure S12, the authors should provide the model/map fit for every TM (1-14)

Response: We agree with the reviewer and have included a new supplemental figure (S13) that includes model/map fits for all 14 transmembrane helices.

Response to the Reviewers

Reviewer 1

Reviewer: The revised manuscript has been improved significantly. We don't have any concerns about the revised manuscript. It is a nice simulation study on the anion exchanger 1.

Response: We thank the reviewer for their constructive feedback on the manuscript.

Reviewer 2

Reviewer: Thank you for addressing my concerns and correcting errors that mentioned.

Response: We thank the reviewer for their constructive feedback on the manuscript.

Reviewer: In the manuscript, three conformations were reported, i.e. IF1-OF, OF-OF and IF1-IF2. The structures were resolved by DSP cross-linking, which absolutely changed the IF prevalence. However, bicarbonate –one of the substrate was not added in the sample, the authors are hypothesizing that AE1 undergoes OF to IF transition over time without bicarbonate according to answer for comment 20. If so, AE1 can transport chloride only, which is not the truth. Something behind remains unknown.

Response: AE1 mediates the exchange of chloride and bicarbonate in a bidirectional manner, i.e., moving one substrate at a time: on the venous side of the blood vessel, it typically imports Cl^- and exports HCO_3^- , whereas on the arterial side, the direction is reversed (Dash et al., *Ann Biomed Eng.*, 2006). That is, Cl^- can bind to both the IF and OF states, as observed in the MD simulations, and induce the conformational change to the other form. Importantly, previous work has shown that AE1 can still mediate chloride exchange in the absence of exogenous bicarbonate by manipulating external and internal Cl^- concentrations (Knauf et al., *J. Gen. Physiol.*, 1996). In our experiments we used a KCl-based buffer, which can enable AE1 to undergo conformational transitions without added bicarbonate.

Reviewer: In the paper titled Cryo-EM structures of the human band 3 transporter indicate a transport mechanism involving the coupled movement of chloride and bicarbonate ions (ref 20), published on Plos Biology Aug 21, 2024. It reported all the conformations you have and more, i.e. IF-OF-Cl-, IF-IF-Cl-, IF-OF-HCO3-, IF-IF-HCO3-, OF-OF-HCO3-. In addition, all the reported structures were resolved in their native states without cross-linking. What kind of innovation would make this work publishable in Nature Communications?

Response: We appreciate the reviewer's comment regarding the recent cryo-EM study (ref 20). Although that study reported an impressive array of native conformations, it noted that in the observed IF structure (corresponding to our IF1 model), TM10 appears as a disordered loop with no extra density on the intracellular side. Our study distinguishes two distinct IF states. In particular, our IF2 model, although derived using a perturbative approach, exhibits an ordered TM10, which reveals potential anion binding sites on the intracellular side and is supported by our molecular dynamics results. Furthermore, we have performed comprehensive computational analyses, including advanced free energy calculations in apo, substrate-bound, and PIP2-free states, to elucidate the transition pathway between the OF and IF states with MD simulations. This mechanistic insight into AE1's conformational dynamics significantly advances our understanding of its substrate transport process. We note that this is one of the first studies in which the effect of the lipids on the transport is demonstrated both quantitatively and mechanistically in a membrane transporter. Finally, our integrated approach unveils previously unrecognized regulatory roles of PIP₂ in modulating AE1 activity. By combining computational, structural, and functional assays, where the last two informed and guided the computations, we generate mechanistic insights that are robustly supported by experimental data, thus strengthening their physiological relevance. Together, these innovations represent substantial advances over prior work and underscore the significance of our study for the field of transporter biology.

Editor

Reviewer: The first pertains to novelty- pointing out that a paper was published in PLoS that reports similar results to you. This falls under our anti-scooping policy as you submitted <1 month from the publication of that work, however you should still make sure to properly acknowledge this work and compare/contrast your results to this.

Response: In response to the reviewer, we have explained in detail how our work differentiates from the previous publication in major ways. We have also properly acknowledged the previous work and integrated it in our discussion. It is of course also cited in our revision.

Reviewer: The second concern relates to how the results presented in the manuscript harmonize with the literature. Specifically, how does the transporter undergo the OF to IF transition without bicarbonate.

Response: Please see above the detailed response to this point.

Reviewer: Please complete or update the following checklist(s) to verify compliance with our research ethics and data reporting standards. Address all points on the checklist, revising your manuscript in response to the points if needed. The form(s) must be downloaded and completed in Adobe Reader rather than opened in a web browser. Each form must be uploaded as a Related Manuscript file at the time of resubmission. Editorial policy checklist: <https://www.nature.com/documents/nr-editorial-policy-checklist.pdf> Reporting summary: <https://www.nature.com/documents/nr-reporting-summary.pdf>

Response: We have completed and uploaded these forms.

Reviewer: DATA AND CODE AVAILABILITY * All Nature Communications manuscripts must include a “Data Availability” section after the Methods section but before the References. If any of the data can only be shared on request or are subject to restrictions, please specify the reasons and explain how, when, and by whom the data can be accessed. For more information on this policy and a list of examples, see: <https://www.nature.com/documents/nr-data-availability-statements-data-citations.pdf>

Response: This has been done.

Reviewer: Official PDB validation reports must be provided with the resubmission. For cryoEM and X-ray crystallography data please upload the updated map and model files to the figshare repository integrated in our submission system (please see above for the information on figshare).

Response: We have provided the required files.

Response to the Reviewer

Reviewer: Thank you for the authors' feedback. It addresses my concerns properly. The only issue I found is that the cryoEM map of IF-OF provided by the author has very poor density in the region of TM10, which is disordered in IF1. It is unlikely to track the main chain of TM10 based on the cryoEM map. I am wondering how did the author build the atomic model of this region and determine the orientation of R730.

Response: We thank the reviewer for their helpful comment regarding the density and map/model fit around R730 in the IF1 state.

In the manuscript, we intended to highlight the order–disorder transition of TM10, which is well ordered in the IF2 state but appears largely unwound and disordered in IF1. In IF1, the density is well resolved up to residue 727 and again from 751 onwards. Although the density is weaker in the region spanning these residues, low-pass filtering reveals the main-chain directionality up to residue 732, allowing us to trace the backbone confidently. While we acknowledge that the side chain of R730 is not sharply defined, the backbone and the position of adjacent residues, where the density is clearer, support our model for TM10.

The images provided in the new supplementary figure (Fig.S2) in the revised manuscript, show two views of the map model fit at the region in question in OF-IF1. They were generated using a Gaussian filter ($\sigma = 1.2$) and a threshold level of 0.1 in UCSF Chimera. We would also like to note that a slight variation in the orientation of R730 does not impact the central conclusion of the manuscript regarding the TM10 order–disorder transition.

Additionally, we identified another subset of particles representing the IF1–IF1 state (not deposited due to its lower overall resolution), in which the density around R730 appears somewhat more straightforward and further supports the model as presented.

We hope this response clarifies our interpretation, and we appreciate the opportunity to address this point.

The manuscript by Chen and colleagues reported three distinct hAE1 TMD conformations: IF1, IF2 and OF. In IF1 the TM10 helix is unfolded, while in IF2 TM10 is fully ordered. In the OF TM10 is also well-ordered. The inhibitory role of PIP2 on ion transport was investigated. Further, MD simulations were performed to analyze the ion binding effect during conformational transition. However, there are many significant discrepancies that needs to be addressed as outlined in my comments below.

Major

1. “During this mechanism, **the movements of the gate and core domains** carry the substrate from one side of the membrane to the other”.

In elevator type movement, the core domain usually undergoes positional displacement, while the gate domain keeps relatively immobile, rather than both gate and core domains move.

2. “while in IF2, TM10 is fully ordered, with R730 interacting with the **C-terminal** end of the TM3 helix (Fig. 1C).”

It not the C-terminal end but the N-terminal end of TM3. The authors should correct this by looking into the structure carefully.

3. “Multiple different configurations of the AE1 dimer were separated by 3D classification, of which three major configurations were refined to high resolution: **IF/OF at 2.4 Å**, OF/OF at 3.1 Å, and IF1/IF2 configuration at 2.9 Å resolution.”

It should be IF1/OF based on Figure S9. The authors should address it properly to keep the consistency throughout the text.

4. “AE1 monomers in the IF1 state represent **48%**, the OF conformation accounts for **32%**, and the IF2 state represents **20%** of the total particles.”

The authors should address the percentage carefully by calculating correct numbers. Refer to both Figure S9 and Supplementary Table, IF1-OF dimer consists of 666k particles, 109k of OF-OF dimer, 79k of IF1-IF2 dimer. So, the number of IF1 monomer is 745k (666k+79k), accounting for 43.6% of all monomer particles. 51.8% for OF monomer and 4.6% for IF2 monomer.

5. “which is alternately accessible to either side of the membrane in the OF and IF states through the elevator movement of the transporter’s gate and core domain . 25 **The gate domain** undergoes the most notable change, particularly in the region that involves TM10.”

As is known, in the IF bAE1 (ref16) and IF AE2 (ref22), the core domain undergoes the most notable change during conformational transition while the gate domain keeps relatively immobile.

6. “Recent studies have demonstrated the involvement of this residue in coordinating HCO – 3 within the binding pocket of AE1.”

Lack of proper reference.

7. “We also observed a second IF state, IF2, in which **R730 is directed towards the intracellular space** but displaced by only 7 Å compared to the OF states.”

It does not look like that R730 is directed towards the intracellular space to me by looking at the middle image of Figure 1C. Please clarify it.

8. "A comparison of the OF and IF conformations reveals that the transition between the states occurs via rotation of the gate domain around an axis roughly intersecting with the intracellular distal corner (F507)."

Again, it is not the gate domain. Also, the authors stated that "A comparison of the OF and IF conformations" but I did not see the comparison in any of the figures and supporting information. The authors should address this important information in Figure 1. Further, the description on OF and IF transition is not sufficient and unclear.

9. Figure 1. There are several concerns that need to be corrected carefully. 1) About the colors used for two monomers. Dark and light green are too similar to each other. 2) Panel A and B should be rearranged. Cryo-EM maps should go first, followed by models. 3) figure legend "The IF1/OF AE1 dimer viewed from the membrane plane (left) and from the cytoplasm (right)." It is actually not the cytoplasmic side. The left image shows the view from the extracellular side. 4) The 90 ° rotation symbol is in the wrong direction, which is not consistent with the left image. 5) Missing key labels. Regardless of every single TM, the authors should at least label the core and gate domain. 6) in panel C, the left schematic representation of IF1, the authors still presented TM10 as α helix, which is misleading.
10. "Lipid headgroup densities are seen at the intracellular dimer interface on the two sides of the protein, where they bind to patches of positively charged residues 2."

I do not understand what is residues 2. Please clarify.

11. "Several amino acids involved in PIP2 binding are conserved within the SLC4 transporter family, particularly P815 and positively charged residues R602 and K817"

The authors demonstrated that R602 is crucial to PIP2 binding, but they did not show R602 in the Figure 2A and B. Please clarify.

12. "When mutated to histidine, it causes recessive distal renal tubular acidosis. 29 This suggests that the positive charge at position 602 may be essential for stabilizing the AE1-PIP2 interaction and, consequently, the dimer interface."

More evidence is needed to support the statement. Shown in Figure B, K600 is closer to PIP2 than R602. Except for R603, both K600 and R603 are positively charged. The dimeric interaction is mostly contributed by hydrophobic interactions of TM6s from two monomers as interpreted in published AE1, AE2, AE3, NBCe1, NDCBE structures.

13. Figure 2 legend error. "In (D), 50 μ M PIP2 was added to liposomes previously treated with PLC."

Which should be In (F)

14. "To test whether PIP2 affects the stability of AE1, we first checked its possible effects on substrate transport using a proteoliposome-based assay. We reconstituted AE1 into liposomes and measured the uptake of NaHCO₃ and NaI to assess the transport activity for the two of the established AE1 substrates. 30 To determine AE1's affinity for HCO₃⁻ and I⁻, isotopic dilutions of the radiolabeled anion with increasing concentrations of the unlabeled anion were performed (Supplementary Fig. S1A), yielding an EC₅₀ (effective concentration of the unlabeled anion to reduce the uptake of radiolabeled anion by 50%) of 79 nM for HCO₃⁻ and 143 nM for I⁻. We verified AE1-specific uptake by performing the assays in the presence of DIDS (4,4'-Diisothiocyano-2,2'-stilbenedisulfonic acid), a known anion transporter inhibitor. 31 The uptake levels in the presence of DIDS were reduced by half compared to those without the inhibitor (Supplementary Fig. S1B)."

What is the conclusion of the effect of PIP2 on AE1 stability? Is the assay carried out here relevant to AE1's stability?

15. How the authors control the orientation of the transporter in liposome, i.e., inside-in or inside-out? If they can not, how do they determine the inside-in/inside-out ratio?
16. "To gain insight into the binding of anions within the lumen formed between the scaffold domain (SD) and transport domain (TD)"

The authors first used gate and core domain, now turned into SD and TD. Should be correct to keep consistency.

17. The estimated dissociation constants K_d for Cl^- and HCO_3^- (255mM and 22mM for OF, 364mM and 38 for IF) are quite high. Do these values reflect the real physiological conditions or may they be influenced by the specific system used in the simulations?
18. The statements of "AE1 monomers in the IF1 state represent 48%, the OF conformation accounts for 32%, and the IF2 state represents 20% of the total particles." and "The elevated free energy of the IF state could explain its lower prevalence in our and previous structural studies . 17–19" are inconsistent. The authors stated first IF1 state represent 48%, OF 32% and IF2 20%, then stated that IF has lower prevalence in their and previous structural studies. Please correct according to my comment 5.
19. Incomplete Figure 6 legend. Missing E and F figure legend. Figure 6B legend is incorrect.
20. The authors stated that substrate binding can lower the energy barrier from OF to IF and the salt bridge between K743 and the 4- or 5-phosphate groups in PIP2 needs to break before reaching the IF state. In the method section, the AE1 sample is prepared without HCO_3^- and without PIP2 removal. But the authors can still get IF1 and IF2 by adding 2.5 mM DSP (dithiobis(succinimidyl propionate)) crosslinker. My question is how can the DSP increasing the prevalence of IF states, where there is no HCO_3^- which means high energy barrier between OF-IF, and salt-bridge between PIP2 and K743 present?

Minor

1. Data availability is not accessible. "Maps and raw movies have been deposited at EMDB (IDs: xxx) and EMPIAR (IDs: yyy)."
2. Clash score is too high for corresponding resolutions provided.
3. EMRinger score should be provided for model validation.
4. Numerous grammar errors need to be carefully corrected.
5. Some format issues. For example: "Membranes were washed 3 times with low-salt buffer (10 mM HEPES, 0.1 M KCl, 0.5 mM dithiothreitol (DTT))" and "The sample was then loaded on a glycerol gradient (12-28% glycerol in HEPES 10 mM, NaCl 130 mM and 0.05% Triton X-100, pH 7.5)"
6. Figure S12, the authors should provide the model/map fit for every TM (1-14)